# Test-Time Adaptation Induces Stronger Accuracy and Agreement-on-the-Line

**Eungyeup Kim**[1]     **Mingjie Sun**[1]     **Christina Baek**[1]     **Aditi Raghunathan**[1]     **J. Zico Kolter**[1,2]

[1]Carnegie Mellon University     [2]Bosch Center for AI

{eungyeuk, mingjies, kbaek, raditi, zkolter}@cs.cmu.edu

## Abstract

Recently, Miller et al. [32] and Baek et al. [3] empirically demonstrated strong linear correlations between in-distribution (ID) versus out-of-distribution (OOD) accuracy and agreement. These trends, coined accuracy-on-the-line (ACL) and agreement-on-the-line (AGL), enable OOD model selection and performance estimation without labeled data. However, these phenomena also break for certain shifts, such as CIFAR10-C Gaussian Noise, posing a critical bottleneck. In this paper, we make a key finding that recent test-time adaptation (TTA) methods not only improve OOD performance, but *drastically strengthen the ACL and AGL trends in models*, even in shifts where models showed very weak correlations before. To analyze this, we revisit the theoretical conditions from Miller et al. [32] that outline the types of distribution shifts needed for perfect ACL in linear models. Surprisingly, these conditions are satisfied after applying TTA to deep models in the penultimate feature embedding space. In particular, TTA causes the data distribution to collapse complex shifts into those can be expressed by a singular "scaling" variable in the feature space. Our results show that by combining TTA with AGL-based estimation methods, we can estimate the OOD performance of models with high precision for a broader set of distribution shifts. This lends us a simple system for selecting the best hyperparameters and adaptation strategy without *any* OOD labeled data. Code is available at `https://github.com/EungyeupKim/TTALine`.

## 1   Introduction

Neural networks often fail to generalize to out-of-distribution (OOD) data that differs from the in-distribution (ID) data seen at train-time [1, 11, 41]. Thus, characterizing the behaviors of these models under distribution shift becomes crucial for reliable deployment. However, it is often extremely challenging to reliably estimate their performances because in many practical applications, OOD labeled data is scarce. Interestingly, recent studies [32, 3] have found a set of simple empirical laws that describe the behavior of models across many distribution shift benchmarks. In particular, across numerous distribution shift benchmarks, the models' ID versus OOD accuracies, under probit scaling, tend to observe a strong linear correlation across numerous distribution shift benchmarks. Additionally, when accuracy is strongly correlated, the ID and OOD agreement rates between pairs of these models are also strongly correlated with nearly identical slopes and biases. These phenomena, respectively referred to as "accuracy-on-the-line" (ACL) [32] and "agreement-on-the-line" (AGL) [3], can be leveraged for precise OOD accuracy estimation *without access to OOD labels*: one could estimate the slope and bias of the ID vs OOD accuracy trend using agreement rates, then linearly transform ID accuracy using this approximate linear fit[1].

---

[1]Throughout this paper, we use AGL as comprehensive term for indicate when models show strong linear trends in both accuracy and agreement, and the slopes and biases of these trends are identical.

38th Conference on Neural Information Processing Systems (NeurIPS 2024).

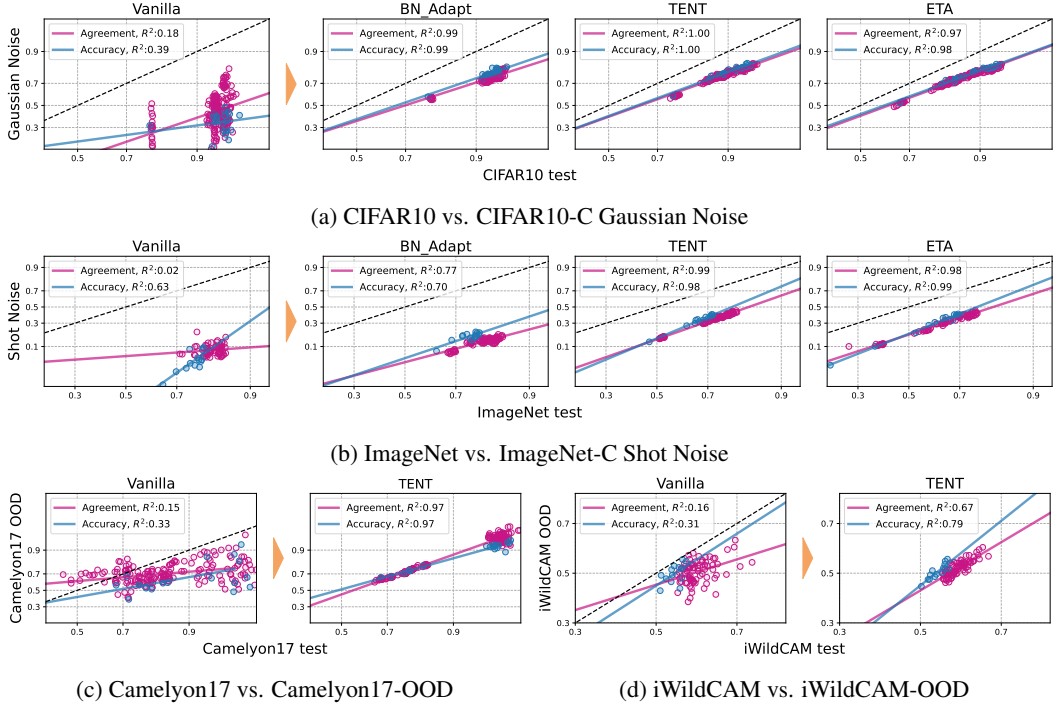

Figure 1: Linear trends in both accuracy and agreement hold to a substantially stronger degree after applying adaptation methods than before. Each blue and pink dot denotes the accuracy and agreement, followed by the linear fits for each, and $R^2$ is correlation coefficient.

However, studies [32, 3, 56, 48] have demonstrated that for distribution shifts benchmarks the linear trends breaks down catastrophically. As shown in Fig. 1, such as CIFAR10-C Gaussian Noise [14] or Camelyon17-WILDS [41], models without TTA (*i.e.*, Vanilla) with around $92 − 95\%$ ID accuracy can have OOD accuracies that vary between $10 − 50\%$, so ID accuracy alone becomes extremely unreliable for understanding the OOD performance of models. Similarly, the correlation strength of ID and OOD agreement rates also weakens. Ideally, we would like to intervene in models in a way that improves these linear correlations, such that we can reliably predict their performance under distribution shift. While theoretical works [32, 31, 51, 25] provide insights for *when* these linear trends hold, there is little study on how to *strengthen* these trends in models.

In this paper, we empirically demonstrate that recent OOD test-time adaptation (TTA) strategies [44, 27, 46, 53, 9, 35, 55, 36] not only improve OOD performance, but significantly *restore the strong linear trends for a broad range of distribution shifts* where ACL and AGL are not initially observable. For instance, in Fig. 1, the correlation coefficient ($R^2$) of ID versus OOD agreement and accuracy of Vanilla models is $0.18$ and $0.39$, respectively, and both of these values improve to $1.0$ after applying TENT [53]. We observe such stronger linear trends after TTA throughout our extensive testbed consisting of 9 shifts, 7 adaptation methods, and over 40 network architectures. As test-time adapted models have OOD accuracies that degrade more predictably with respect to the ID accuracy, we are also able to utilize AGL based method ALine [3] to obtain precise OOD performance estimates without any OOD labels. Note that *no OOD labels were utilized throughout this procedure*, neither during TTA or ALine. Our estimates of the models' OOD performances are drastically more precise after adaptation, *e.g.*, estimation error of $11.99\%$ in Vanilla models without TTA vs. $2.34\%$ after applying TENT on CIFAR10-C Gaussian Noise.

Given that TTAs are designed to enhance OOD accuracy, the observation of such strong linear trends is unexpected and non-trivial. While some studies [29, 9] have explored how adaptations lead to improved OOD generalization, these efforts are orthogonal to those investigating the conditions under which linear trends occur [32, 31]. To our knowledge, no studies have attempted to bridge these two areas of research. This naturally raises the question: Why does adapting models at test-time to OOD data lead to stronger linear trends?

To answer this, we revisit the theoretical analysis in Miller et al. [32] of the sufficient conditions for observing highly correlated ID vs OOD accuracy in linear classifiers and Gaussian data. Theoretically, ACL holds exactly under distribution shifts where the direction of the class means and shape of the class covariances are fixed, and only the scale of the mean or covariance changes. Surprisingly, we find that TTA, in practice, seems to enforce exactly this condition: after applying TTA, the cosine similarity between the means and covariances of the penultimate-layer feature embedding of ID and OOD data tend to hover around 1, while their magnitude may change by some scaling constant. This implies that adaptations effectively collapse the complexity of the distribution shift to a singular "scaling" variable in the feature space. In addition to (empirically) justifying the use of TTA for strengthening ACL[2], this discovery casts some insight into the nature of TTA in general, which has previously been a largely heuristic approach.

Furthermore, models adapted with different TTA hyperparameters tend to lie on the *same* linear trend. Fig. 2 illustrates the strong correlation among models of various architectures first trained on ID data for various numbers of epochs, then adapted with different learning rates, batch sizes, adaptation steps. This critically allows us to tune the TTA hyperparameters and choice of TTA method without a held-out OOD labeled set. Notably, correctly hyperparameter tuning without access to labels is an important challenge faced in practice that currently lacks principled approaches [21, 60, 9]. Using ACL and AGL, we are able to select models with accuracy less than $1\%$ away from that of the best model OOD, *e.g.*, in CIFAR10-C. This also allows to select the best TTA methods by comparing their estimated best OOD performances.

To summarize our contributions:

- We observe after TTA, ACL and AGL hold across a wider set of distribution shifts and hyperparameter settings.

- We explain our observation by showing TTA collapses the distribution shift to just a constant scaling of the mean and covariance matrices in the feature space. This satisfies the theoretical conditions studied previously for observing strong linear trends.

- Our findings provide a simple and effective strategy for finding the best TTA hyperparameters and the best TTA strategy without any OOD labels.

## 2 Related Work

**Understanding accuracy and agreement-on-the-line.** Miller et al. [32] and Baek et al. [3] empirically observed a coupled phenomena in deep models when evaluated on many standard distribution shift benchmarks: the ID vs. OOD accuracy and agreement are often strongly correlated and the linear fits match almost exactly. Recent studies [32, 31, 51, 25, 24] attempt to characterize what types of shifts lead to (or break) this phenomena. Miller et al. [32] prove that under a simple Gaussian data setup, ACL does not hold perfectly under distribution shifts that change the direction of the mean or transforms the covariance matrix. For example, they demonstrate that adding isotropic Gaussian noise to CIFAR10 whose covariance is anisotropic, causes the linear trend to break. Mania and Sra [31] provided sufficient conditions directly over the outputs of trained models, in terms of their prediction similarity and distributional closeness. Tripuraneni et al. [51] and LeJeune et al. [25] show that ACL holds asymptotically under certain transformations to the covariance matrix. On the other hand, there has been comparatively little theoretical analysis of AGL and why it appears together with ACL. Lee et al. [24] show that in random feature linear regression, AGL can break break partially, *i.e.*, the slope of the agreement trend matches accuracy's, but the biases may be different. While further theoretical conditions are necessary to guarantee when these phenomena hold jointly, in our empirical findings, we see that the slopes and biases do always match across the wide variety of distribution shifts we test.

Extending upon such theoretical studies, we demonstrate that a simple model intervention by test-time adaption allows these ID versus OOD trends to hold even stronger for a more expansive set of distribution shifts. In fact, we demonstrate that after adaptation, models actually satisfy the theoretical conditions necessary for ACL as described in Miller et al. [32].

---

[2]AGL has not been similarly shown to hold theoretically for this case of scaled means, but empirically it often seems to hold when ACL holds [3]

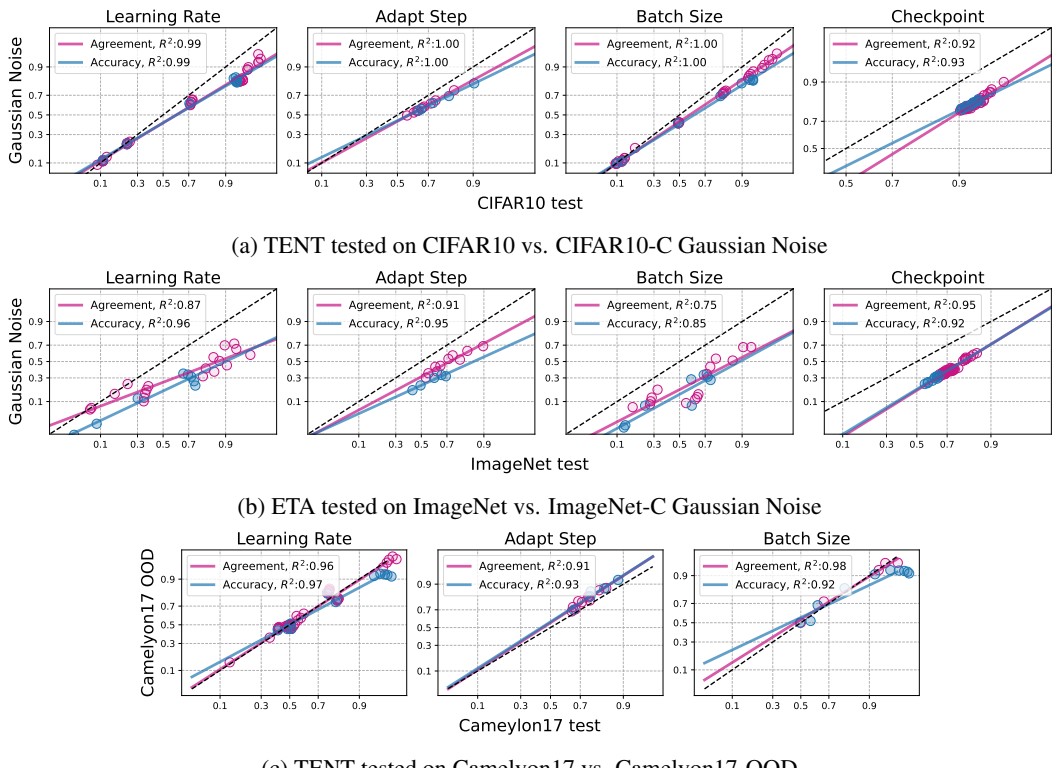

(a) TENT tested on CIFAR10 vs. CIFAR10-C Gaussian Noise

(b) ETA tested on ImageNet vs. ImageNet-C Gaussian Noise

(c) TENT tested on Camelyon17 vs. Camelyon17-OOD

Figure 2: Strong AGL by adaptations with varying hyperparameters, including learning rates, adaptation steps, batch sizes, and early-stopped checkpoints of the ID-trained model. Each blue and pink dot denotes the accuracy and agreement, followed by the linear fits for each.

**Adaptations under distribution shifts and their pitfalls of reliability.** Test-time adaptation aims to enhance model robustness by adapting models to unlabeled OOD test data. Test-time training methods [46, 29, 7] involve learning shift-invariant features via solving self-supervision tasks. Other approaches include computing statistics in Batch Normalization (BN) layers using OOD data [20, 44], instead of using ID statistics stored during training. Subsequent studies further adapt BN parameters by updating them using entropy minimization [53, 35, 36]. Another popular approach is self-training with pseudo-labels [40, 54, 9].

One critical challenge in TTA is that, without OOD labels, it is prohibitively difficult to evaluate how effective the adaptation methods might be. As pointed out in previous studies, adaptation methods may not succeed to address the full spectrum of distribution shifts, such as datasets reproductions [29, 60], domain generalization benchmarks [21, 61], and WILDS [40]. Furthermore, these approaches are known to be sensitive to different hyperparameter choices [4, 61, 36, 23]. Practitioners must take a great care in optimizing such hyperparameters, but their tuning procedures lack clarity. They often follow the settings of the previous studies [35, 36], or rely on some held-out labeled data [21, 60, 9] that is unavailable in practice. There exists a line of studies on unsupervised model validation [33, 42, 34, 52, 17]. Our study leverages *agreement-on-the-line phenomena within TTA* to offer a promising solution for these reliability issues.

# 3 Adaptations lead to stronger Agreement-on-the-Line

## 3.1 Experimental setup

**Datasets and models.** Our testbed includes diverse shifts, including common corruptions (15 failure shifts in CIFAR10-C, CIFAR100-C, and ImageNet-C [14]), dataset reproductions (CIFAR10.1 [38], ImageNetV2 [39]), and real-world shifts (ImageNet-R [16], Camelyon17-WILDS, iWildCAM-WILDS, FMoW-WILDS [41]). Among them, CIFAR10-C Gaussian Noise, Camelyon17-WILDS,

and iWildCAM-WILDS display the weakest correlations in both ID versus OOD accuracy and agreement [32, 3, 56]. We test over 30 different architectures of convolutional neural networks (*e.g.*, VGG [45], ResNet [13, 57, 59], DenseNet [19], MobileNet [43]) and Vision-Transformers (ViTs [5], DeiT [49], SwinT [30]).

**TTA methods.** We investigate 7 recent state-of-the-art TTA methods, including SHOT [27], BN_Adapt [44], TTT [46], TENT [53], ConjPL [9], ETA [35], and SAR [36]. This testbed includes different training strategies (*e.g.*, self-supervision [46], PolyLoss [26]) and updating certain layer parameters (BN layers [44, 53, 35], LayerNorm (LN) layers [2, 36], entire feature extractors [27]). We test all adaptation baselines, except SAR, on convolutional neural networks with BN layers. We apply SAR specifically to vision transformers [5, 49, 30] because it prevents the model collapse that other adaptation methods exhibit in vision transformers with LN layers [36].

**Calculating agreement.** Given any pair of models $(h, h') \in \mathcal{H}$ that are tested on distribution $\mathcal{D}$, the expected accuracy and agreement of the models is defined as

$$\text{Accuracy}(h) = \mathbb{E}_{x,y\sim\mathcal{D}}\big[\mathbb{1}\{h(x) = y\}\big], \quad \text{Agreement}(h, h') = \mathbb{E}_{x,y\sim\mathcal{D}}\big[\mathbb{1}\{h(x) = h'(x)\}\big] \quad (1)$$

where $h(x)$ and $h'(x)$ are the normalized logits of models $h$ and $h'$ given datapoint $x$ and $y$ is the class label. Following [32] and [3], we apply probit scaling, which is the inverse of the cumulative density function of the standard Gaussian distribution ($\Phi^{-1} : [0, 1] \to [-\infty, \infty]$), over accuracy and agreement for a better linear fit, specifically in Figs. 1 and 2.

**Online test in ID and OOD during TTA.** Unlike conventional offline inference at test-time, TTAs involve online learning, which dynamically updates the model's parameters while testing on OOD data. To evaluate this continuously updated model on both ID and OOD data, the data is fed into the model in minibatches and we collect the accuracy and agreement of the model over the $i$'th minibatch after the dynamic adaptation step $i$. Details are provided in Algorithm 1 in the Appendix.

## 3.2 Main observation

We make a striking empirical observation in the models after applying TTA: the correlation strength between ID versus OOD accuracy and agreement increases by a significant margin, especially in distribution shifts where models without TTA show wildly varying trends. In Fig. 1, we demonstrate this on the four distribution shifts that show weakest ACL and AGL trends in Vanilla models: CIFAR10-C Gaussian Noise, ImageNet-C Shot Noise, Camelon17-WILDS, and iWildCAM-WILDS. In particular, the correlation coefficients are at most $R^2 \leq 0.4$ before TTA. These failure shifts have also been noted by previous studies [32, 3, 56]. Surprisingly, after applying TTA methods, such as TENT, the strength of these linear trends as measured by $R^2$ increase dramatically, (*e.g.*, $0.15 \to 0.97$ and $0.33 \to 0.97$ in agreement and accuracy in Camelyon17-WILDS). Our observations hold consistently across our entire testbed of shifts and adaptation methods, as shown in Figs. 6, 7, 9, 10, 11, and 12. Moreover, in shifts where models without TTA already exhibit strong linear trends, such as ImageNet-V2 or FMoW-WILDS, these linear trends persist even after TTA, irrespective of whether TTA improves or degrades the OOD accuracy (Fig. 4).

Furthermore, models adapted using the same TTA method but with different adaptation hyperparameters lie on the same accuracy and agreement linear trend. Specifically, we vary learning rates, the number of adaption steps, batch sizes, and the number of epochs the model is initially trained on ID data. In Fig. 2 we see that on CIFAR10-C and ImageNet-C Gaussian Noise, and Camelyon17-WILDS, models adapted with different hyperparameters exhibit correlation strength $R^2$ close to 1 in both ID versus OOD accuracy and agreement. We show that this occurs across distribution shifts in Figs. 8 and 13.

## 4  Why does adaptations lead to strong linear trends?

In this section, we focus on explaining why TTA leads to restoration of these strong linear trends. Although we do not provide a complete theoretical justification, we identify a key pattern in how TTA modifies models that provides a strong clue as to why these methods substantially strengthen ACL and AGL.

We begin by revisiting the theoretically sufficient conditions for ACL from Miller et al. [32] over a simple Gaussian data and linear classifier setup. In this setting, ACL holds perfectly ($R^2 = 1$) only

when the distribution shifts by a simple scaling factor to the norm of the mean and covariance. On the other hand, if the actual direction of the mean or shape of the covariance matrix changes, the linearity breaks. We then demonstrate that even in CIFAR10-C shifts and deep neural networks, models after applying TTA meet these theoretical conditions better in the penultimate-layer feature space.

## 4.1 Theoretical conditions for linear trends in Gaussian data

We briefly introduce the theoretical conditions for ACL in Gaussian data and linear models, from Miller et al. [32], restated here for ease of presentation. Consider the binary classification over Gaussian data setup with label $y \in \{-1, 1\}$. The OOD distribution $Q$ differs from the ID distribution $P$ by just some scaling constants $\alpha, \gamma > 0$,

$$P(x \mid y) = \mathcal{N}(y \cdot \mu; \Sigma), \quad Q(x \mid y) = \mathcal{N}\left(y \cdot \alpha\mu; \gamma^2\Sigma\right). \tag{2}$$

**Theorem 1** *[Miller et al. [32], simplified] Under the Gaussian data setup in Equation 2, across all linear classifiers $f_\theta : x \mapsto \text{sign}(\theta^\top x)$, the probit-scaled accuracies over P and Q observes perfect linear correlation with a bias of zero and a slope of $\frac{\alpha}{\gamma}$.*

The proof follows immediately from the fact that the test accuracy of a linear classifier on this Gaussian data $P$ is given by $\Phi(\theta^\top \mu / \sqrt{\theta^\top \Sigma \theta}))$; applying same result to $Q$ immediately gives the desired linear relationship. The main implication is that if the distribution shifts only in the scale of the mean and covariance, while preserving their direction, ACL is guaranteed to occur across any set of linear classifiers. Note that we focus on the conditions for ACL in particular. Additional theoretical conditions are usually required to observe AGL together with ACL [24], but they have not been well-characterized especially for this linear-Gaussian setup. Thus, we forgo analyzing assumptions for AGL, if any, in our work. Still, we find that AGL is always tightly coupled with ACL across an extensive range of benchmarks and TTA methods, similar to Baek et al. [3].

## 4.2 Empirical analysis under adaptation to CIFAR10-C

We now investigate how well the above conditions are met before and after TTA for real-world data with deep models. Here, evaluate models trained on CIFAR10 on CIFAR10-C Gaussian Noise. We adapt models using BN_Adapt [44] or TENT [53] at test-time. To match the theoretical setup, we analyze the class-wise feature embeddings from the penultimate-layer of these models that directly precedes the final *linear* classifier. Namely, we can measure the "alignment" or the cosine similarity between the ID and OOD class means and covariances of these embeddings. Similar to our CIFAR10-C Gaussian Noise results in Figs. 1 and 2, we evaluate the embedding alignment of models in each hyperparameter setup: we adapt models at test-time varying a single hyperparameter, e.g., architectures, learning rates, batch sizes, and early-stopped checkpoints, while keeping the remaining hyperparameters fixed.

**Shifts become aligned in their mean and covariance after adaptation.** Table 1 shows the mean and standard deviation of the cosine similarity measured across different setups. We first notice that, without adaptation (Vanilla), both mean and covariance of CIFAR10 and CIFAR10-C Gaussian Noise features are misaligned with cosine similarity much smaller than 1. Surprisingly, after adaptations, the similarity substantially increases and becomes very close to 1, across every architecture we test (with standard deviation close to 0). In other words, in the feature embedding space, the two distributions have means and covariances that (roughly) point in the same *directions* after adaptation. This also implies that in adapted models, the distribution shift is represented as a simple change in the *scale* of the features alone.

Consider CIFAR10-C Gaussian Noise which corrupts CIFAR10 images by additive isotropic Gaussian Noise. In the input image space, the direction of the covariance matrix clearly changes with the addition of this noise. Yet TTA has the non-trivial ability of being able to "simplify" complex drifts to scale shifts within the features. These observations support the theoretical conditions in Theorem 1 and gives us loose insight into why TTA strengthens ACL and AGL in models in practice.

**Theoretical slope matches the empirical slope.** Finally, under scale shifts in our theoretical setup, the slope of the ACL trend is exactly $\Delta = \alpha/\gamma$. Assuming the direction of the mean and covariance remains fixed, we may estimate $\alpha$ and $\gamma$ as simply the average change in magnitude of the class means and covariances, and use this to approximate $\Delta$. In Table 1 we compare how this theoretical

| | Cosine Similarity | | Slope | |
| Setup | Mean | Covariance | Theoretical | Empirical |
|---|---|---|---|---|
| Vanilla (Archs.) | $0.691 \pm 0.175$ | $0.750 \pm 0.109$ | – | – |
| BN_Adapt (Archs.) | $0.988 \pm 0.007$ | $0.972 \pm 0.011$ | $0.751 \pm 0.075$ | 0.758 |
| TENT (Archs.) | $0.990 \pm 0.005$ | $0.974 \pm 0.011$ | $0.753 \pm 0.072$ | 0.778 |
| Learning rates | $0.993 \pm 0.003$ | $0.977 \pm 0.006$ | $0.759 \pm 0.041$ | 0.76 |
| Batch Sizes | $0.995 \pm 0.003$ | $0.982 \pm 0.010$ | $0.831 \pm 0.101$ | 0.809 |
| Check Points | $0.992 \pm 0.003$ | $0.976 \pm 0.008$ | $0.782 \pm 0.033$ | 0.838 |

Table 1: Cosine similarity between mean direction and covariance shape of class-wise penultimate-layer features, followed by the comparison between theoretical and empirical slope. They are evaluated on CIFAR10 vs. CIFAR10-C Gaussian Noise, measured across architectures and hyperparameters. We report their means and standard deviations.

slope compares to the true slope of ID versus OOD accuracy fit by linear regression. These slopes tend to closely match each other, highlighting that the theoretical results of TTA seem to provide correct intuition about why ACL holds in practice, especially after TTA.

**TTA improves linear trends irrespective of whether it improves OOD generalization.** We make the subtle distiction between TTA's ability to strengthen ACL and AGL phenomena from its original purpose of improving OOD accuracy. One could imagine a trivial reason why ACL happens is because the ID and OOD gap diminishes after TTA, thereby bringing the ID versus OOD accuracy of all models close to the $y = x$ line. However, this explanation does not reflect our empirical findings. While TTA seems to reduce the complexity of the shift to scale shifts, the magnitude of this scaling factor may remain very large, $i.e.$, $\alpha \ll 1$, $\gamma \gg 1$. This results in a near-perfect linear trend that, nonetheless, lies arbitrarily far from the $y = x$. Consider the shift ImageNet-C Shot Noise in Fig. 1. After adaptation, there is still approximately a 40% gap in ID and OOD accuracy, but ACL holds strongly. On the other hand, consider the counterfactual TTA method which improves OOD performance by decreasing the covariance scale $\gamma$ in the feature embedding space, but the means/covariances remain misaligned. Then the linear trend moves closer to $y = x$, but there is no guarantee about the strength of the linear trend.

# 5 Experiments

Our finding that TTA induces strong ACL and AGL has two key practical applications: (i) estimation of OOD accuracy, and (ii) unsupervised validation for TTA — all without access to any OOD labels.

## 5.1 OOD Accuracy estimation after adaptation

**Experimental Setup.** We employ AGL-based estimation method, namely ALine [3]. This method first estimates the slope and bias of the linear trend in ID versus OOD accuracy, via agreements (which requires no labels), and uses them to linearly transform ID accuracy to get an estimate of OOD accuracy. The full details of ALine are illustrated in Algorithm 2 in Appendix. We apply ALine to models after TTA, and compare these results with (i) those of ALine applied to models before adaptation, and (ii) estimates using existing estimation baselines. These baselines include average thresholded confidence (ATC) [8], difference of confidence (DOC)-feat [10], average confidence [15], and agreement [22]. We evaluate baselines on CIFAR10-C, CIFAR100-C, ImageNet-C [14], and Camelyon17-WILDS and iWildCAM-WILDS [41].

**Results.** ALine shows accurate estimation under shifts where models demonstrate strong AGL ($e.g.$, in CIFAR10-C snow, mean absolute error (MAE) of estimation is 0.93% in ALine-D), but critically fails when shifts do not have such linear trends ($e.g.$, in CIFAR10-C Gaussian Noise, MAE is 10.76% and in Camelyon17-WILDS, MAE is 12.88%). As a result, as shown in Table 2, estimates using ALine-S/D on Vanilla models are sometimes error-prone. Next, after applying adaptation methods, such as SHOT, BN_Adapt, and TENT, the estimation performance of ALine-S/D improves, showing substantially lower MAE compared to that of Vanilla models ($e.g.$, MAE decreases $10.76\% \rightarrow 2.34\%$ in CIFAR10-C Gaussian Noise and $12.88\% \rightarrow 1.42\%$ in Camelyon17-WILDS). ALine on top of TTA also outperforms other baseline methods on most of the distribution shifts tested.

| Dataset | Method | Error | ATC | DOC-feat | AC | Agreement | ALine-S | ALine-D |
|---------|--------|-------|-----|----------|-----|-----------|---------|---------|
| CIFAR10-C | Vanilla | 31.38 | 8.31 | 15.03 | 17.42 | 5.45 | 6.02 | 5.87 |
| | SHOT | 15.40 | 1.63 | 4.63 | 7.63 | 1.78 | 0.96 | **0.77** |
| | BN_Adapt | 16.87 | 3.69 | 4.79 | 7.53 | 1.93 | 1.12 | **0.91** |
| | TENT | 15.43 | 4.25 | 4.65 | 7.66 | 1.79 | 0.97 | **0.77** |
| | ConjPL | 16.62 | 1.80 | 6.16 | 11.46 | 2.02 | 1.18 | **1.01** |
| | ETA | 15.14 | 4.58 | 4.50 | 7.68 | 1.76 | 0.92 | **0.72** |
| CIFAR100-C | Vanilla | 59.04 | 5.05 | 12.82 | 18.34 | 6.96 | 7.49 | 7.22 |
| | SHOT | 40.79 | 2.21 | 5.44 | 14.36 | 2.52 | 1.64 | **0.90** |
| | BN_Adapt | 42.69 | 2.89 | 4.42 | 11.81 | 2.33 | 1.43 | **1.13** |
| | TENT | 41.11 | 6.60 | 5.59 | 14.85 | 2.65 | 1.64 | **0.88** |
| | ConjPL | 42.79 | **1.09** | 6.55 | 23.73 | 2.40 | 1.67 | 1.18 |
| | ETA | 44.27 | 7.15 | 4.92 | 16.49 | 4.96 | 1.44 | **0.81** |
| ImageNet-C | Vanilla | 80.41 | 3.95 | 13.72 | 17.34 | 9.06 | 6.00 | 5.95 |
| | BN_Adapt | 69.05 | 7.37 | **2.63** | 2.86 | 3.91 | 6.16 | 6.09 |
| | TENT | 56.58 | 5.98 | 6.54 | 12.70 | 7.48 | 4.62 | **4.57** |
| | ETA | 56.56 | 10.21 | 7.91 | 34.38 | 8.02 | **3.66** | 3.72 |
| | SAR | 43.30 | 5.39 | 8.61 | 13.68 | 5.51 | 5.19 | **4.17** |
| Camelyon17 -WILDS | Vanilla | 34.07 | 14.91 | 17.31 | 21.69 | 11.95 | 12.88 | 13.46 |
| | TENT | 14.37 | 3.00 | 3.43 | 6.94 | 6.49 | 2.29 | **2.27** |
| | ETA | 16.43 | 3.05 | 4.38 | 6.85 | 5.33 | 2.24 | **1.42** |
| iWildCAM -WILDS | Vanilla | 50.27 | 7.12 | 2.73 | 23.86 | 3.00 | 3.53 | 2.82 |
| | TENT | 47.39 | 5.44 | 3.20 | 28.03 | 3.55 | **2.59** | 2.96 |
| | ETA | 46.49 | 6.61 | 3.40 | 29.34 | 4.62 | **2.14** | 2.82 |

Table 2: The results of OOD accuracy estimation, measured by MAE (%) between estimated and actual OOD accuracy. The gray shades denote the results calculated after applying adaptations, and bold texts indicate the smallest estimation error among estimators. We also report the classification error (%) in both Vanilla and adaptation methods for each dataset.

## 5.2 Unsupervised validation for TTA

In this section, we demonstrate that strong AGL allows us to validate the best hyperparameter for TTA without any labels. Furthermore, we can use the OOD accuracy estimates to choose the best TTA strategy *overall*.

**Experimental setup.** First, as we have shown, models adapted using a particular method with varying hyperparameters tend to show strong ACL behavior. When ACL holds, we can simply select the best OOD model by selecting the one with best ID accuracy [32]. Thus, we first collect candidates by systematically sweeping over hyperparameter values, and select the model with the best ID accuracy. We focus on TENT, optimizing its various hyperparameters including learning rates, adaptation steps, architectures, batch sizes, and early-stopped checkpoints of training. See our total pool of models in Appendix Table 6. We compare this naive strategy to existing unsupervised validation methods including MixVal [17], ENT [33], IM [34], Corr-C [52], and SND [42]. We evaluate them on four shifts, CIFAR10-C, ImageNet-C, ImageNet-R, and Camelyon17-WILDS.

**Results.** Table 3 reports the absolute difference (MAE (%)) between the OOD accuracy of the selected model and the best OOD accuracy, within a set of models where we vary a particular TTA hyperparameter. We find that using ID accuracy for model selection consistently achieves competitive unsupervised validation results, often outperforming other baselines. We noticed that current state-of-the-art unsupervised model selection methods, *i.e.*, MixVal or IM, perform well on CIFAR10-C, ImageNet-C, and ImageNet-R, but they critically fail in Camelyon17-WILDS, *e.g.*, validation error of 7.98% in MixVal and 23.52% in IM. Such failures of existing baselines might come from their assumptions, *e.g.*, low-density separation, that do not generalize to such distribution shifts. In contrast, our method shows low MAE across all distribution shifts, including those where other baselines fail, *e.g.*, 0.62% in Camelyon17-WILDS. We note that there are cases, specifically models adapted with various learning rates for ImageNet-C, where MAE is large (9.70%). We see that models optimized with very small learning rates, *e.g.*, $10^{-5}$, deviate from the linear trend. Results on other TTA baselines' unsupervised validation are in Table 7.

**Extension to selecting best TTA strategy.** After hyperparameter tuning for each TTA strategy using the unsupervised strategy from the previous section, we could further leverage ALine to then select for the best TTA method overall. Different TTA methods exhibit different slopes and biases in their ACL

| HyperParameter | CIFAR10-C | | | | | | ImageNet-C | | | | | |
|---|---|---|---|---|---|---|---|---|---|---|---|---|
| | MixVal | ENT | IM | Corr-C | SND | Ours | MixVal | ENT | IM | Corr-C | SND | Ours |
| Architecture | 2.31 | 1.06 | 1.06 | 21.71 | 2.77 | 0.03 | 6.22 | 0.96 | 0.47 | 26.32 | 20.60 | 0.75 |
| Learning Rate | 6.97 | 8.88 | 2.24 | 11.56 | 1.87 | 0.72 | 12.75 | 20.49 | 1.49 | 20.18 | 12.61 | 9.70 |
| Checkpoints | 3.21 | 0.0 | 0.0 | 5.53 | 3.46 | 0.05 | – | – | – | – | – | – |
| Batch Size | 7.85 | 3.32 | 0.96 | 32.37 | 5.68 | 0.77 | 14.29 | 42.31 | 0.99 | 42.31 | 42.31 | 5.61 |
| Adapt Step | 0.85 | 0.0 | 0.0 | 1.02 | 0.0 | 0.23 | 1.85 | 1.94 | 1.25 | 3.09 | 2.17 | 0.30 |
| Average | 4.23 | 2.65 | 0.85 | 14.43 | 2.75 | **0.36** | 8.77 | 16.42 | **1.05** | 14.43 | 22.97 | 4.0 |

| HyperParameter | ImageNet-R | | | | | | Camelyon17-WILDS | | | | | |
|---|---|---|---|---|---|---|---|---|---|---|---|---|
| | MixVal | ENT | IM | Corr-C | SND | Ours | MixVal | ENT | IM | Corr-C | SND | Ours |
| Architecture | 1.75 | 0.62 | 0.62 | 22.17 | 22.17 | 0.85 | 28.87 | 1.03 | 1.03 | 28.87 | 28.87 | 0.85 |
| Learning Rate | 3.12 | 10.16 | 4.73 | 19.16 | 19.16 | 2.8 | 0.91 | 48.37 | 46.41 | 48.37 | 48.37 | 1.14 |
| Batch Size | 1.83 | 35.88 | 0.08 | 35.88 | 35.88 | 1.74 | 0.0 | 46.67 | 46.67 | 40.45 | 40.45 | 1.37 |
| Adapt Step | 1.07 | 1.07 | 1.07 | 1.07 | 1.07 | 0.0 | 2.17 | 33.12 | 0.0 | 33.12 | 33.12 | 0.0 |
| Average | 1.94 | 14.18 | 1.62 | 19.57 | 19.57 | **1.34** | 7.98 | 32.29 | 23.52 | 37.70 | 37.70 | **0.62** |

Table 3: Results of unsupervised validation, measured by MAE (%) between OOD accuracy of model selected by best ID model and actual best OOD model. Our results are highlighted in shade.

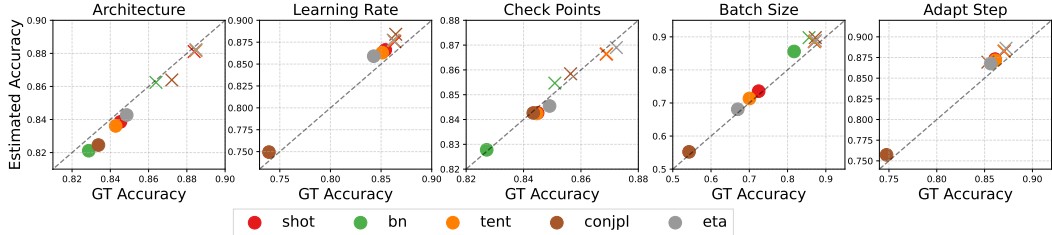

Figure 3: 2-D visualizations of each TTA baseline's ground truth best OOD accuracy (x-axis) and estimated OOD accuracy of best-ID models (y-axis) marked in cross (×). Each color denotes different TTA baselines. Circle dot (∘) represents estimates and ground truth accuracies averaged over all hyperparameter values.

trends, so we use AGL-based estimators to estimate the best OOD accuracy across hyperparameters for each method separately, then compare these estimates to choose the best method. Specifically, we select between five TTA baselines — BN_Adapt, TENT, SHOT, ConjPL, and ETA — on CIFAR10-C across all corruptions. For ease of presentation, we consider selecting the best TTA method after tuning a single hyperparameter for each method, with other parameters fixed. In Fig. 3, we plot using "×" the selected models' estimated OOD accuracy for each method versus each method's actual best OOD performances. Our method almost always selects the best OOD model for each method but also accurately estimates their performances, as shown by points lying close to the $y = x$ line. Furthermore, the ranking of TTA methods by OOD accuracy stays the same when using the estimates. We also plot in "∘" the average ground truth and estimated accuracy across hyperparameters. This demonstrates that our approach effectively selects the best TTA strategy without OOD labels.

# 6 Ablation Study: When does TTA not improve linear trends?

So far, we focused on numerous TTA methods that tend to dramatically improve the strength of ACL and AGL phenomena. We then connected this behavior to models representing the distribution shift in its feature space as a scale shift, satisfying the theoretical conditions of ACL from Miller et al. [32]. This behavior was especially prominent in methods that adjust the batch normalization layer statistics, *e.g.* BN_Adapt, or modify the feature extractor, *e.g.* TENT. This section presents an ablation study on the impact of normalization layers (*e.g.*, BN, LN) and the components adapted by TTA (*e.g.*, feature extractor, final classifier) for observing ACL and AGL.

**Normalization layers.**

As shown in our results for BN_Adapt in Section 4.1, the strength of the ACL and AGL trends tends to be sensitive to the statistics stored in the batch normalization layers. Taking the same set of models

from Section 4.1, we test two architectural variants — models with no normalization layers (denoted as Vanilla w/o N) and layer normalization (Vanilla w/ LN) — and report the cosine similarity of the class means and covariance matrices of the ID and OOD feature embeddings.

We also report the correlation coefficient ($R^2$) of the ACL and AGL trends in Table 4. Interestingly, the mean and covariance alignment as well as correlation strength tends to be larger than that of Vanilla with BN-layers, but still falls short of models with BN-layers adapted with BN_Adapt which is very close to 1.

Methods such as SAR which substitutes BN-layers with LN as a part of their adaptation strategy, also shows stronger linear trends in Fig. 12 than Vanilla counterparts in Fig. 11, but not as

| Setup | Cosine Similarity | | Correlations | |
|---|---|---|---|---|
| | Mean | Covariance | Acc. | Agr. |
| **Vanilla w/o N** | $0.794 \pm 0.117$ | $0.778 \pm 0.172$ | 0.29 | 0.71 |
| **Vanilla w/ LN** | $0.935 \pm 0.042$ | $0.854 \pm 0.063$ | 0.56 | 0.86 |
| **Vanilla w/ BN** | $0.691 \pm 0.175$ | $0.750 \pm 0.109$ | 0.18 | 0.39 |
| **BN_Adapt** | $0.998 \pm 0.007$ | $0.972 \pm 0.011$ | 0.99 | 0.99 |

Table 4: Cosine similarity of mean direction / covariance shape and correlation coefficients ($R^2$) in CIFAR10-C Gaussian Noise, measured across architectures. Last two rows are from Table 1.

consistently as those of BN-layer-adaptation methods in Fig. 11. We conjecture that this stems from how different architectures encode distribution shifts: BN-layers result in severe covariate shift, but most of the shift is encoded in the BN statistics, making it easy to align shifts via adaptation. In contrast, non-BN models encode shifts across the entire model parameters, which may dampen the covariate shift in the representations, but also make it harder to remove by adaptation. Schneider et al. [44] suggests a similar conclusion.

**Featurizer or classifier.** In this section, we consider the TTA method T3A [21], which leverages a feature extractor without BN layers and only adapts the last linear classifier. This is in contrast to the other TTA methods we study in our work, which largely modifies the feature extractor. We adapt the same set of models in Section 4.1 and observe that the $R^2$ for accuracy and agreement in T3A are $0.80$ and $0.46$, which is substantially lower than models adapted using BN_Adapt. This may be because the features remain misaligned after T3A. Overall, our ablation studies clarify what TTA designs ideally lead to strong linear trends. Since non-BN-TTAs are often adopted in numerous practical circumstances, *e.g.*, single batch, it remains an important open question how to develop alternate TTA strategies that achieves strong ACL and AGL trends in such settings, thus improving the reliability of models.

# 7   Conclusion and Limitations

In this paper, we provide a key observation that recent TTAs lead to stronger AGL across a wide range of distribution shifts, encompassing those with weak correlations before adaptation. We explain this phenomena by the complexity of distribution shifts being substantially reduced to those where the direction/shape of the mean and covariance remains identical, satisfying the theoretical conditions for linear trends. This naturally leads to enhanced estimation of OOD performances across a wider range of shifts than before TTA. We can leverage this to perform unsupervised hyperparameter validation and select for the best TTA method without any labels.

While the method we propose does not require OOD labels, AGL still requires sufficient ID data to accurately estimate agreement rate. As a result, our strategy may not be complementary with TTA methods that do not require any ID data. Moreover, it could make our strategy computationally expensive. In Section D, we conduct an ablation study to minimize the required amount of ID data for observing AGL for accuracy estimation. The results show that even with only 5% of the original ID data, OOD accuracy estimation performance remains nearly the same as with full access, outperforming other estimators. We believe that overcoming dependency on ID data and exploring a fully test-time approach for observing AGL remains a promising direction for future research.

# Acknowledgement

We acknowledge the anonymous reviewers for their valuable feedbacks. Eungyeup Kim, Mingjie Sun, and Christina Baek are supported by funding from the Bosch Center for Artificial Intelligence. Aditi Raghunathan gratefully acknowledges support from Open Philanthropy, Google, Apple and Schmidt AI2050 Early Career Fellowship.

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

# A Details on Experimental Setup

## A.1 Model Architectures

We list the types of architectures for each testbed of dataset.

| CIFAR10, 100 Architectures | ImageNet Architectures | WILDS Architectures |
|---|---|---|
| ResNet18 [13] | ResNet18 [13] | ResNet18 [13] |
| ResNet26 [13] | ResNet34 [13] | ResNet34 [13] |
| ResNet34 [13] | ResNet50 [13] | ResNet50 [13] |
| ResNet50 [13] | ResNet101 [13] | ResNet101 [13] |
| ResNet101 [13] | ResNet152 [13] | ResNet152 [13] |
| WideResNet28 [59] | ResNeXT32×4d [57] | ResNeXT32×4d [57] |
| PreActResNet18 [12] | DenseNet121 [19] | DenseNet121 [19] |
| PreActResNet34 [12] | WideResNet50 [59] | WideResNet50 [59] |
| PreActResNet50 [12] | WideResNet101 [59] | WideResNet101 [59] |
| PreActResNet101 [12] | DLA34 [58] | VGG11 [45] |
| RegNet X200 [37] | DLA46C [58] | VGG13 [45] |
| RegNet X400 [37] | DLA60 [58] | VGG16 [45] |
| RegNet Y400 [37] | DLA102 [58] | VGG19 [45] |
| VGG11 [45] | DLA169 [58] | |
| VGG13 [45] | ViT-B/16 [5] | |
| VGG16 [45] | ViT-L/16 [5] | |
| VGG19 [45] | ViT-B/32 [5] | |
| MobileNetV2 [43] | SwinT-S [30] | |
| PNASNet-A [28] | SwinT-B [30] | |
| PNASNet-B [28] | SwinT-L [30] | |
| SENet18 [18] | DeiT-S/16 [49] | |
| GoogLeNet [47] | DeiT-B/16 [49] | |
| | DeiT3-B/16 [50] | |
| | DeiT3-L/16 [50] | |
| | DeiT3-H/14 [50] | |

Table 5: The list of architecture types for each testbed of datasets, including CIFAR10, CIFAR100, ImageNet, Camelyon17-WILDS, and iWildCAM-WILDS.

For CIFAR10, CIFAR100, and WILDS datasets, we train the models from the scratch, while for ImageNet, we use the pretrained model weights from torchvision and timm package. In addition, since TTT requires specific network composition required for the rotation-prediction task during pretraining, we train them using ResNet-14,26,32,50,104, and 152, which are available in the original implementation[3].

For Camelyon17 dataset, we also test the neural networks with their feature extractor (all parameters before the final linear classifier) randomly initialized, following Miller et al. [32] that tested the low-accuracy models on the dataset. We notice that these models, even with their most of parameters being randomized, still achieve high accuracy in both ID and OOD after training the last linear classifier. They also exhibit improved OOD accuracy after adaptations. We apply this randomization to ResNet18, ResNet34, ResNet50, VGG16, VGG13, and VGG11.

We trained and tested all models and datasets in NVIDIA RTX 6000.

## A.2 Distribution Shifts

We test the models on 9 different distribution shifts that include synthetic corruptions and real-world shifts. Here are details of tested benchmarks:

---

[3]https://github.com/yueatsprograms/ttt_cifar_release

**Synthetic corruptions.** CIFAR10-C, CIFAR100-C, and ImageNet-C [14] are designed to apply the 15 different types of corruptions, such as Gaussian Noise, on their original dataset counterparts. We use the most severe corruptions, which have severity of 5, in all experiments. These corruptions datasets are most commonly evaluated distribution shifts in a wide range of TTA papers [44, 46, 29, 27, 53, 9, 35, 36, 61].

**Dataset reproduction.** CIFAR10.1 and ImageNetV2 are reproduced version of their original counterparts, CIFAR10 and ImageNet, respectively. They are collected by carefully following the dataset creation procedure of the original dataset, yet resulting in non-trivial performance degradations to models trained on original counterparts.

**Style shifts.** ImageNet-R is one variant of ImageNet which contains the images with renditions of various styles. Such styles include paintings, cartoons, sketches and others, which induces significant distribution shifts even when the semantics are invariant.

**Real-world shifts.** Camelyon17-WILDS [41] contains the medical images of tissue collected from different hospitals, where distribution shifts are designed by different hospitals the model is trained and tested. The task is to predict whether it has tumor tissue or not (binary classification). We followed the same evaluation protocol of Miller et al. [32], collecting 30 whole-slide images (WSI) from 3 hospitals as ID (total 302,436 for train and 33,560 for test), while 10 WSI from a different hospital as OOD (total 85,054). FMoW-WILDS [41] contains the spatio-temporal satellite imagery of 62 different use of land or building categories, where distribution shifts originate from the years that the imagery is taken. Specifically, following Miller et al. [32], we use ID set consists of images taken from 2002 to 2013 (training data of 76,863 and test of 11,327), and OOD set taken between 2013 and 2016 (test data of 22,108). iWildCAM-WILDS [41] contains the images of animals of 182 total species, which are taken with different camera traps. Following Miller et al. [32], we use 129,809 train images and 8,154 test images taken by the same 243 camera traps as ID. OOD data consists of total 42,791 images, which are taken with different camera traps, which involve different camera angle, background, and others.

## A.3 Adaptation hyperparameters

Table 6 shows the hyperparameter pools for each setup when used for observing AGL across different hyperparameters. We use SGD optimizer with momentum of 0.9 for all adaptation baselines except for SAR, which uses sharpness-aware minimization (SAM) optimizer [6].

| Setup | CIFAR10, CIFAR100 | ImageNet |
|---|---|---|
| Learning Rates | $1 \cdot 10^{-5}, 5 \cdot 10^{-5}, 1 \cdot 10^{-4}, 5 \cdot 10^{-4}, 1 \cdot 10^{-3},$ $5 \cdot 10^{-3}, 1 \cdot 10^{-2}, 5 \cdot 10^{-2}, 1 \cdot 10^{-1}, 5 \cdot 10^{-1}$ | $1 \cdot 10^{-5}, 2 \cdot 10^{-5}, 5 \cdot 10^{-5}, 1 \cdot 10^{-4}, 2 \cdot 10^{-4},$ $5 \cdot 10^{-4}, 1 \cdot 10^{-3}, 2 \cdot 10^{-3}, 5 \cdot 10^{-3}$ |
| Batch Sizes | $1, 2, 4, 8, 16, 32, 64, 128, 256$ | $4, 8, 16, 32, 64, 128, 256, 512$ |
| Early-stopped Checkpoints | $100, 110, 120, 130, 140, 150, 160, 170, 180, 190$ (out of 200 total epochs) | – |
| Adaptation Steps | $1, 2, 3, 4, 5$ | $1, 2, 3, 4, 5$ |

(a) CIFAR10, CIFAR100, ImageNet

| Setup | Camelyon17-WILDS | iWildCAM-WILDS |
|---|---|---|
| Learning Rates | $2 \cdot 10^{-5}, 5 \cdot 10^{-5}, 1 \cdot 10^{-4}, 2 \cdot 10^{-4}, 5 \cdot 10^{-4}, 1 \cdot 10^{-3},$ $2 \cdot 10^{-3}, 5 \cdot 10^{-3}, 1 \cdot 10^{-2}, 2 \cdot 10^{-2}, 5 \cdot 10^{-2}$ | $2 \cdot 10^{-5}, 5 \cdot 10^{-5}, 1 \cdot 10^{-4}, 2 \cdot 10^{-4},$ $5 \cdot 10^{-4}, 1 \cdot 10^{-3}, 2 \cdot 10^{-3}, 5 \cdot 10^{-3}$ |
| Batch Sizes | $4, 8, 16, 32, 64, 128, 256, 512$ | $1, 2, 4, 8, 16, 32, 64, 128$ |
| Early-stopped Checkpoints | – | – |
| Adaptation Steps | $1, 2, 3, 4, 5$ | $1, 2, 3, 4, 5$ |

(b) Camelyon17-WILDS, iWildCAM-WILDS

Table 6: The hyperparameter pools utilized for observing AGL across hyperparameters in CIFAR10, CIFAR100, ImageNet, Camelyon17-WILDS, and iWildCAM-WILDS dataset.

## A.4 Online test in ID and OOD during TTA

Algorithm 1 provides the details of how ID and OOD performances are tested during TTA. TTA updates the model parameters at each iteration using the different batch of unlabeled OOD data ($x_{\text{OOD}}$), as described in L6 of Algorithm 1. Note that during TTA, we also provide a batch of ID data ($x_{\text{ID}}$). After adaptation, we inference both ID and OOD data to the updated model to obtain its predictions (L7-8), stored for performance evaluation at the end of test. Stored predictions are used for calculating not just accuracy, but agreements which require additional model.

Such online test is different from traditional testing. Traditional testing tests the model that is fixed in its pre-trained parameters, while the online "test" is testing a model that is continuously updated its parameters during TTA. In other words, the model tested at iteration of $t - 1$ is different from that at iteration $t$. Specifically in Algorithm 1, L6 makes the only difference between traditional and online testing, where L6 updates the model parameters every iteration before testing in L7 and L8.

---

**Algorithm 1** Online Test in ID and OOD during TTA

---
1: **Inputs:** A pair of data $\mathcal{X}_{\text{ID}}, \mathcal{Y}_{\text{ID}} \mathcal{X}_{\text{OOD}}, \mathcal{Y}_{\text{OOD}}$ model $h_\theta$.
2: **Algorithms:** Adaptation objective $\mathcal{L}_{\text{TTA}}(\cdot)$.
3: ─────────────────────────
4: Prediction sets $\mathcal{P}_{\text{ID}} = \emptyset, \mathcal{P}_{\text{OOD}} = \emptyset$
5: **for** batch $(x_{\text{ID}}, y_{\text{ID}}), (x_{\text{OOD}}, y_{\text{OOD}})$ in $\mathcal{X}_{\text{ID}}, \mathcal{Y}_{\text{ID}}, \mathcal{X}_{\text{OOD}}, \mathcal{Y}_{\text{OOD}}$ **do**
6:      $\theta \leftarrow \theta - \eta \nabla \mathcal{L}_{\text{TTA}}(h_\theta(x_{\text{OOD}}))$                 ▷ Apply TTA
7:      $\mathcal{P}_{\text{ID}} = \mathcal{P}_{\text{ID}} \cup h_\theta(x_{\text{ID}})$
8:      $\mathcal{P}_{\text{OOD}} = \mathcal{P}_{\text{OOD}} \cup h_\theta(x_{\text{OOD}})$
9: **end for**
10: **return** $\text{Acc}(\mathcal{P}_{\text{ID}}, \mathcal{Y}_{\text{ID}}), \text{Acc}(\mathcal{P}_{\text{OOD}}, \mathcal{Y}_{\text{OOD}})$

---

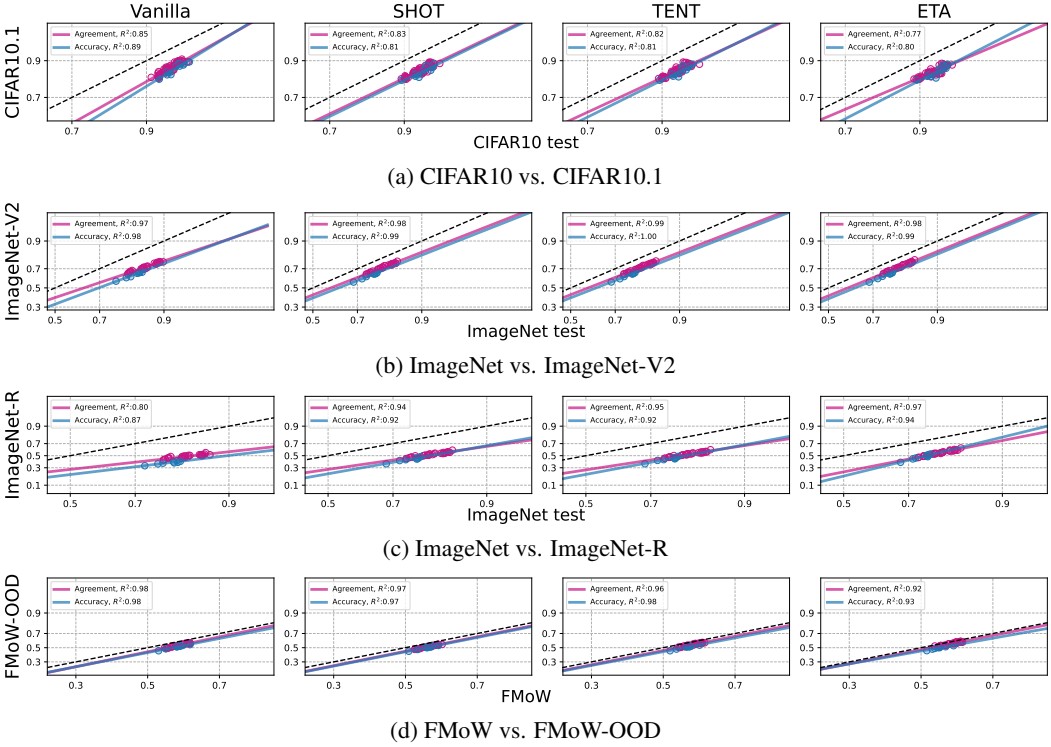

Figure 4: The linear trends visualization of shifts that already exhibit strong AGL in Vanilla. Notice that adaptations do not break the linear trends, even when they lead to accuracy drops. We test SHOT, TENT, and ETA on shifts such as CIFAR10.1, ImageNetV2, ImageNet-R, and FMoW-WILDS. Each blue and pink dot denotes the accuracy and agreement, followed by the linear fits for each. The axes are probit scaled.

## B  Analysis on distribution shifts that have AGL without adaptations

We also examine the distribution shifts that exhibit strong AGL even before TTA, which are dataset reproductions (CIFAR10.1 [38], ImageNetV2 [39]), and other real-world shifts (ImageNet-R [16], FMoW-WILDS [41]), as shown in Fig. 4. The results show that adaptation baselines, such as SHOT, TENT, and ETA, do not improve OOD generalizations or even degrade them, which were also evidenced by previous studies [53, 61]. However, they persist to have the strong linear trends, which implies that TTA does not break the linearity in shifts that show the strong correlations. This suggests that we can reliably predict if TTA may succeed or falter in a wide range of distribution shifts.

## C  OOD accuracy estimation baselines

### C.1  ALine-S and ALine-D

Baek et al. [3] propose ALine-S and ALine-D, which assess the models' OOD accuracy without access to labels by leveraging the agreement-on-the-line among models. We provide the detailed algorithm of ALine-S and ALine-D in Algorithm 2.

### C.2  Average thresholded confidence (ATC)

Garg et al. [8] introduce OOD accuracy estimation method, ATC, which learns the confidence threshold and predicts the OOD accuracy by using the fraction of unlabeled OOD samples for which model's negative entropy is less that threshold. Specifically, let $h(x) \in \mathbb{R}^c$ denote the softmax output of model $h$ given data $x$ from $\mathcal{X}_{\text{OOD}}$ for classifying among $c$ classes. The method can be written as

**Algorithm 2** ALine-S and ALine-D

---

1: **Input:** ID predictions $\mathcal{P}_{\text{ID}}$ and labels $\mathcal{Y}_{\text{ID}}$, OOD predictions $\mathcal{P}_{\text{OOD}}$.
2: **Function:** Probit transform $\Phi^{-1}(\cdot)$, Linear regression $\mathcal{F}(\cdot)$.

---

4: $\hat{a}, \hat{b} = \mathcal{F}(\Phi^{-1}(\text{Agr}(\mathcal{P}_{\text{ID}})), \Phi^{-1}(\text{Agr}(\mathcal{P}_{\text{OOD}})))$      ▷ Estimate slope and bias of linear fit
5: $\widehat{\text{Acc}}_{\text{OOD}}^{\text{S}} = \Phi(\hat{a} \cdot \text{Acc}(\mathcal{P}_{\text{ID}}, y_{\text{ID}}) + \hat{b})$      ▷ ALine-S
6: Initialize $A \in \mathbb{R}^{\frac{n(n-1)}{2} \times n}, b = \mathbb{R}^{\frac{n(n-1)}{2}}$
7: i=0
8: **for** $(p_{j,\text{ID}}, p_{k,\text{ID}}), (p_{j,\text{OOD}}, p_{k,\text{OOD}}) \in \mathcal{P}_{\text{ID}}, \mathcal{P}_{\text{OOD}}$ **do**
9:     $A_{ij} = \frac{1}{2}, A_{ik} = \frac{1}{2}, A_{il} = 0 \forall l \notin j, k$
10:    $b_i = \Phi^{-1}(\text{Agr}(p_{j,\text{OOD}}, p_{k,\text{OOD}})) + \hat{a} \cdot \left( \frac{\Phi^{-1}(\text{Acc}(p_{j,\text{ID}}, y_{\text{ID}})) + \Phi^{-1}(\text{Acc}(p_{k,\text{ID}}, y_{\text{ID}}))}{2} - \Phi^{-1}(\text{Agr}(p_{j,\text{ID}}, p_{k,\text{ID}})) \right)$
11:     i=i+1
12: **end for**
13: $w^* = \arg\min_{w \in \mathbb{R}^n} ||Aw - b||_2^2$
14: $\widehat{\text{Acc}}_{\text{OOD}}^{\text{D}} = \Phi(w_i^*) \forall i \in [n]$      ▷ ALine-D
15: **return** $\widehat{\text{Acc}}_{\text{OOD}}^{\text{S}}, \widehat{\text{Acc}}_{\text{OOD}}^{\text{D}}$

---

below:

$$\widehat{\text{Acc}}_{\text{OOD}} = \mathbb{E}\big[\mathbb{1}\{s(h(x)) < t\}\big], \tag{3}$$

where $s$ is the negative entropy, *i.e.*, $s(h(x)) = \sum_c h_c(x) \log(h_c(x))$, and $t$ satisfies

$$\mathbb{E}\big[\mathbb{1}\{s(h(x)) < t\}\big] = \mathbb{E}\big[\mathbb{1}\{\arg\max_c h_c(x) \neq y\}\big]. \tag{4}$$

### C.3 Difference of confidence (DOC)-feat

Guillory et al. [10] observe that the shift of distributions is encoded in the difference of model's confidences between them. Based on this observation, they leverage such differences in confidences as the accuracy gap under distribution shifts for calculating the final OOD accuracy. Specifically,

$$\widehat{\text{Acc}}_{\text{OOD}} = \text{Acc}_{\text{ID}} - \left( \mathbb{E}\big[\max_c h_c(x_{\text{ID}})\big] - \mathbb{E}\big[\max_c h_c(x_{\text{OOD}})\big] \right). \tag{5}$$

### C.4 Average confidence (AC)

Hendrycks and Gimpel [15] estimate the OOD accuracy based on model's averaged confidence, which can be written as

$$\widehat{\text{Acc}}_{\text{OOD}} = \mathbb{E}\big[\max_c h(x_{\text{OOD}})\big]. \tag{6}$$

### C.5 Agreement

Jiang et al. [22] observe that disagreement between the models that are trained with different setups closely tracks the error of models in ID. We adopt this as the baseline for assessing generalization under distribution shifts, where we can estimate $\widehat{\text{Acc}}_{\text{OOD}} = \text{Agr}(\mathcal{P}_{\text{OOD}})$, where $\mathcal{P}_{\text{OOD}}$ denotes the set of predictions of the models on OOD data $\mathcal{X}_{\text{OOD}}$.

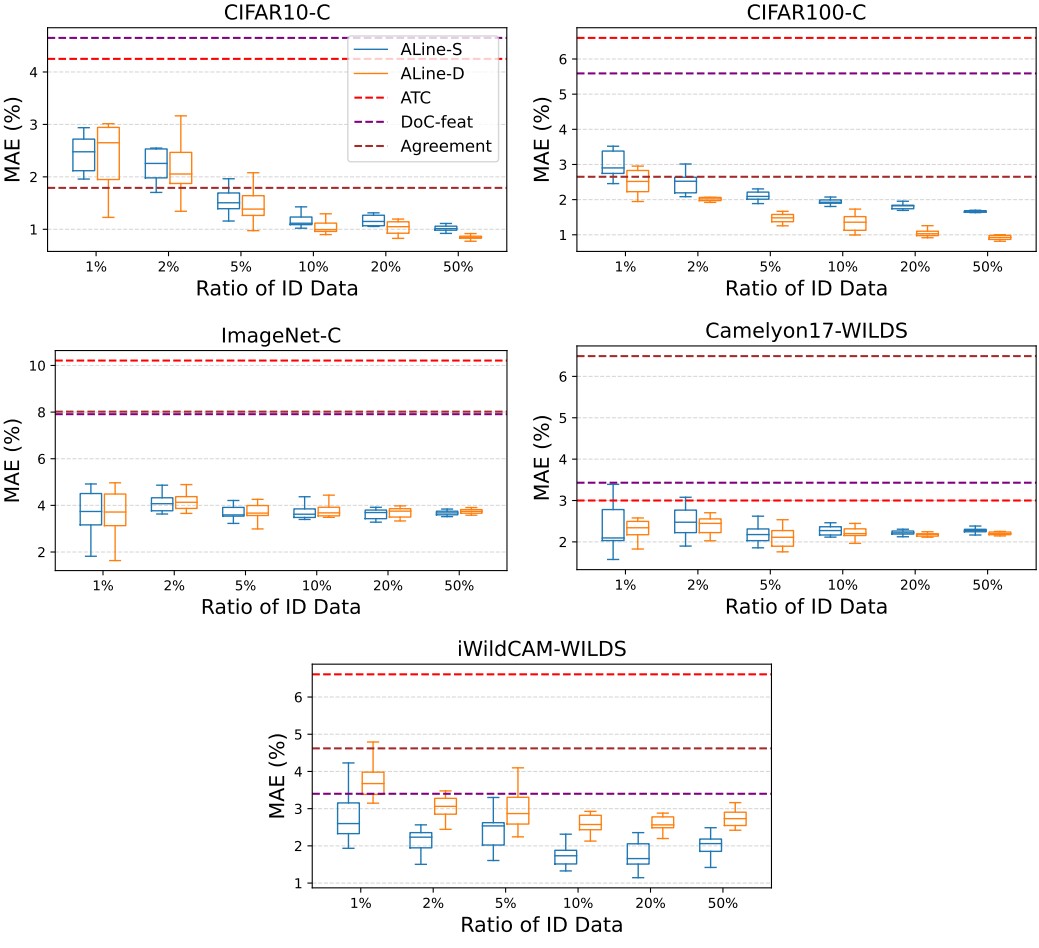

Figure 5: OOD accuracy estimation results with limited amount of ID data, decreasing from $50\%$ to $1\%$ of entire data pool. We randomly sampled 10 different subsets for each ratios, and visualize the distribution of MAE (%) results. The results of baseline including ATC, DoC-feat, and Agreement are included for comparison.

# D   Ablation study on the number of ID data

Even if access to labeled ID data is often easily available in practice more than obtaining OOD data, ID data could be limited in certain applications due to privacy issue or computational complexity. Therefore, this section further explores how sensitive our accuracy estimations are under conditions where the number of ID samples is critically limited.

Specifically, for each dataset, we gradually decrease the ratio of ID data for calculating both accuracy and agreement in ID, ranging from $50\%$ to $1\%$ of entire data. To report the confidence interval of different random subsets, we iterate 10 independent sampling of data for each ratio. Box plots in Fig. 5 show that for CIFAR10-C and CIFAR100-C, with even $5\%$ begins to become close to precision of using $10\times$ more, *i.e.*, $50\%$, achieving the state-of-the-art estimation performances outperforming other baselines. Also, in more complex dataset such as ImageNet-C, Camelyon17-WILDS, and iWildCAM-WILDS, even using $1\%$ of entire ID data achieves the best estimation performances among baselines. These results indicate that our framework is practically feasible in such practical applications where access to ID is highly limited.

| HyperParameter | CIFAR10-C | | | | | | CIFAR100-C | | | | |
|---|---|---|---|---|---|---|---|---|---|---|---|
| | BN_Adapt | TTT | SHOT | TENT | ETA | ConjPL | BN_Adapt | SHOT | TENT | ETA | ConjPL |
| Learning Rate | – | 3.71 | 0.65 | 0.72 | 0.72 | 0.42 | – | 0.90 | 1.05 | 0.67 | 0.68 |
| Adapt Step | – | 0.04 | 0.23 | 0.23 | 0.24 | 0.12 | – | 0.63 | 0.0 | 0.55 | 0.63 |
| Architecture | 0.21 | 0.49 | 0.03 | 0.03 | 0.01 | 0.04 | 0.61 | 1.13 | 0.69 | 0.84 | 0.38 |
| Batch Size | 0.0 | – | 0.73 | 0.77 | 0.77 | 0.18 | 0.02 | 0.57 | 0.50 | 0.62 | 0.75 |
| Checkpoints | 0.0 | 0.48 | 0.07 | 0.05 | 0.01 | 0.11 | 0.01 | 0.00 | 0.24 | 0.0 | 0.0 |
| Average | 0.07 | 1.18 | 0.34 | 0.36 | 0.35 | 0.17 | 0.21 | 0.48 | 0.49 | 0.53 | 0.48 |

| HyperParameter | ImageNet-C | | | | ImageNet-R | | | Camelyon17 | | iWildCAM | |
|---|---|---|---|---|---|---|---|---|---|---|---|
| | BN_Adapt | TENT | ETA | SAR | BN_Adapt | TENT | ETA | BN_Adapt | TENT | BN_Adapt | TENT |
| Learning Rate | – | 9.70 | 6.67 | 6.21 | – | 2.80 | 5.50 | – | 1.14 | – | 3.17 |
| Adapt Step | – | 0.30 | 0.0 | 3.07 | – | 0.0 | 0.0 | – | 0.0 | – | 1.97 |
| Architecture | 2.23 | 0.39 | 0.24 | 0.75 | 0.06 | 0.85 | 1.83 | 1.22 | 0.0 | 0.0 | 4.08 |
| Batch Size | 0.00 | 5.61 | 5.70 | 7.21 | 0.0 | 1.74 | 4.02 | 0.0 | 1.37 | 0.15 | 2.04 |
| Average | 1.15 | 4.0 | 3.15 | 4.31 | 0.03 | 1.34 | 2.83 | 0.61 | 1.25 | 0.07 | 2.81 |

Table 7: Results of unsupervised validation of TTA methods other than TENT, measured by MAE (%) between OOD accuracy of model selected by best ID model and actual best OOD model. Since BN_Adapt do not involve parameter updates, we exclude learning rate and the adaptation steps. In addition, TTT uses a single batch, and we exclude the batch size.

# E   Unsupervised validation results on TTA baselines

This section supplements Table 3 by the unsupervised validation results of various TTA methods other than TENT, which include those of BN_Adapt, TTT, SHOT, ETA, ConjPL, and SAR. Table 7 reports the difference in OOD accuracy (MAE (%)) between the model with best ID accuracy and the actual best OOD model, tested across different adaptation hyperparameters. The OOD model selection via best ID model achieves less than 1% MAE over almost every hyperparameter setup and adaptations, in CIFAR10-C and CIFAR100-C every corruptions in average. It also demonstrates that the hyperparameter optimization in ImageNet-C, R and WILDS benchmark datasets have reasonably low MAE across diverse setups and adaptation baselines. These indicate that the strong linear trends across hyperparameters enable the selection of near-optimal OOD adaptations across various shifts and TTA strategies.

# F    Additional results on CIFAR10-C, CIFAR100-C, and ImageNet-C

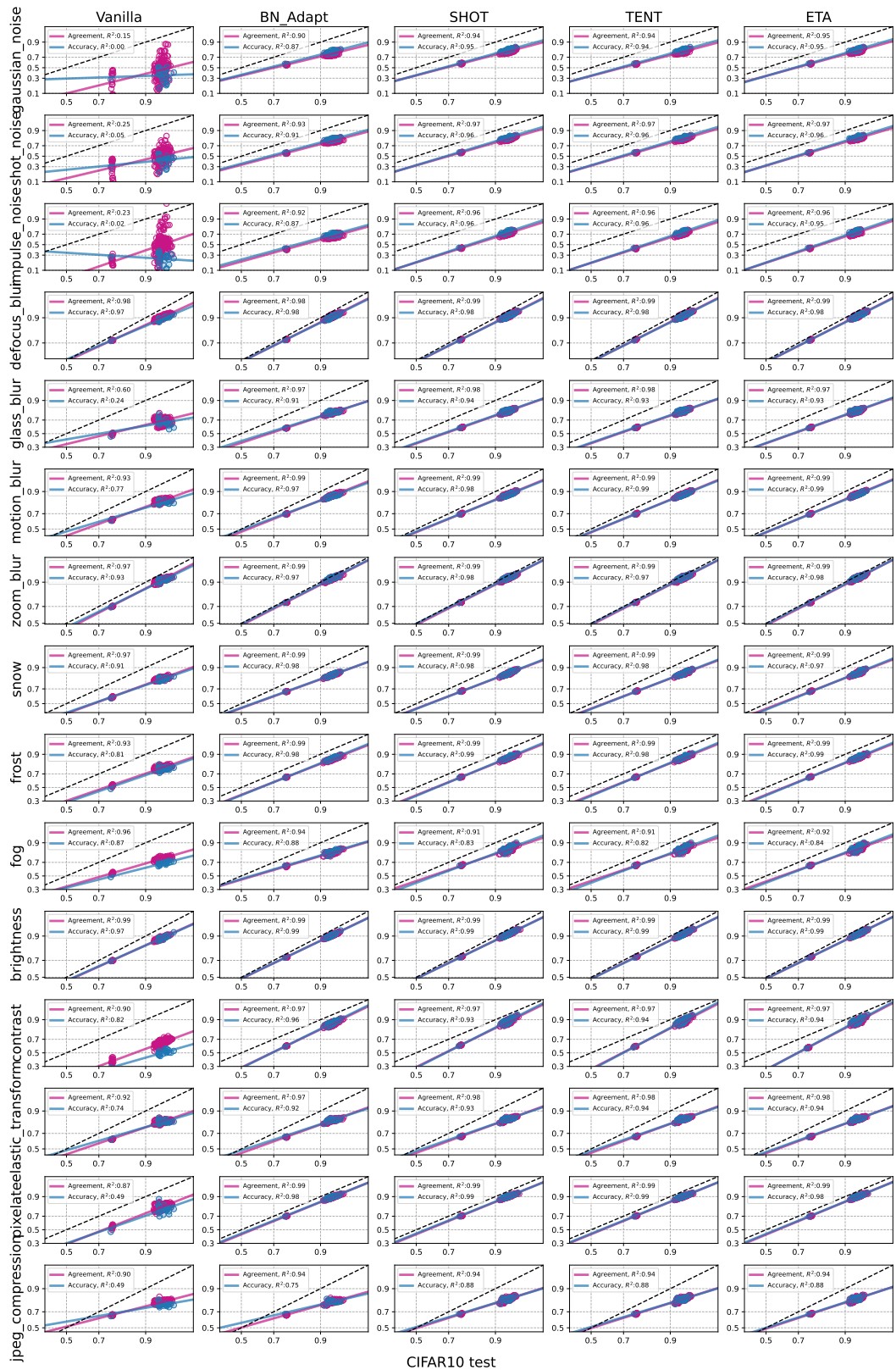

Figure 6: Results on CIFAR10-C corruptions (different architectures), BN_Adapt [44], SHOT [27], TENT [53], and ETA [35].

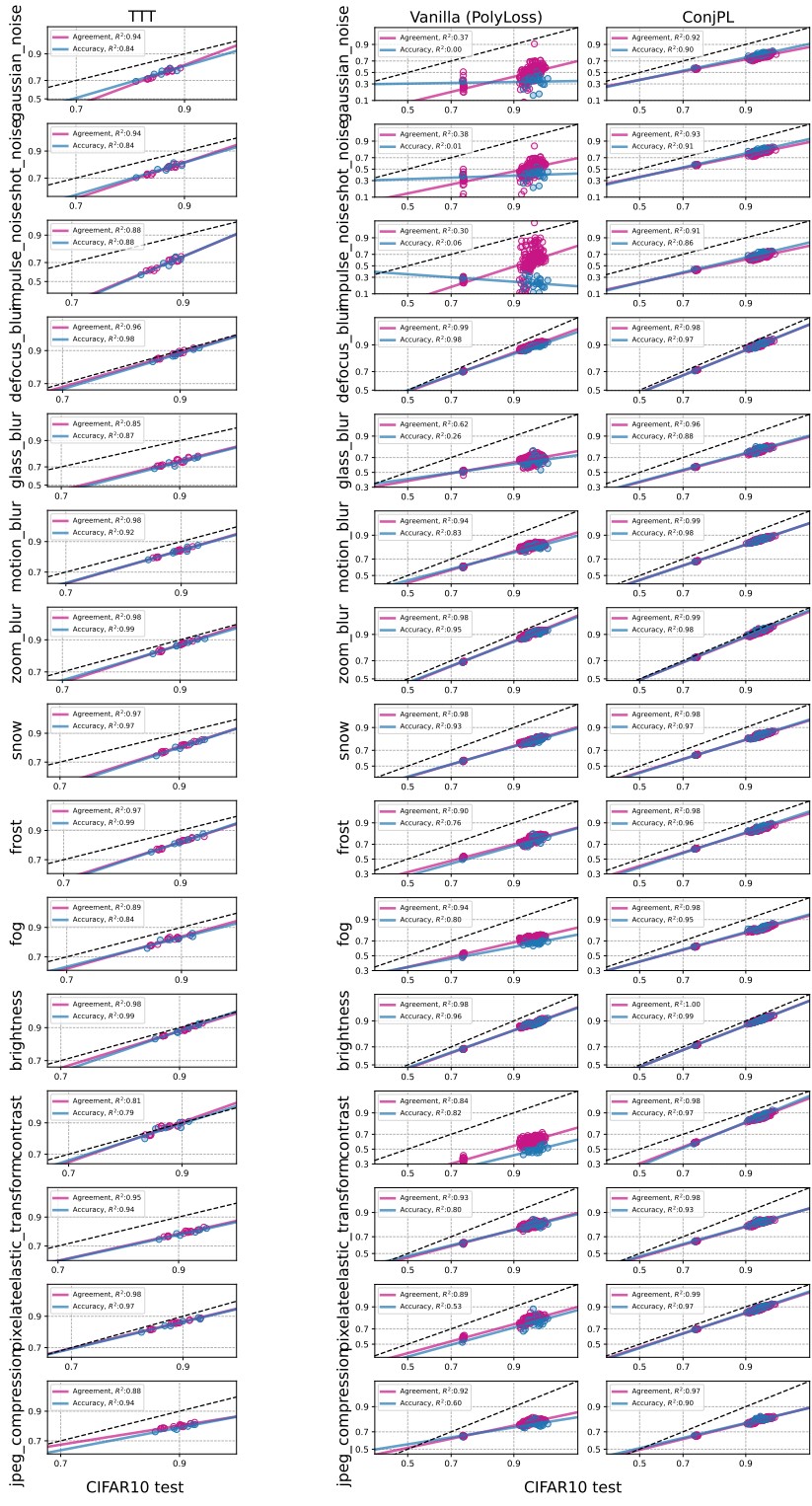

Figure 7: Results on CIFAR10-C corruptions (different architectures), TTT [46], ConjPL [9]

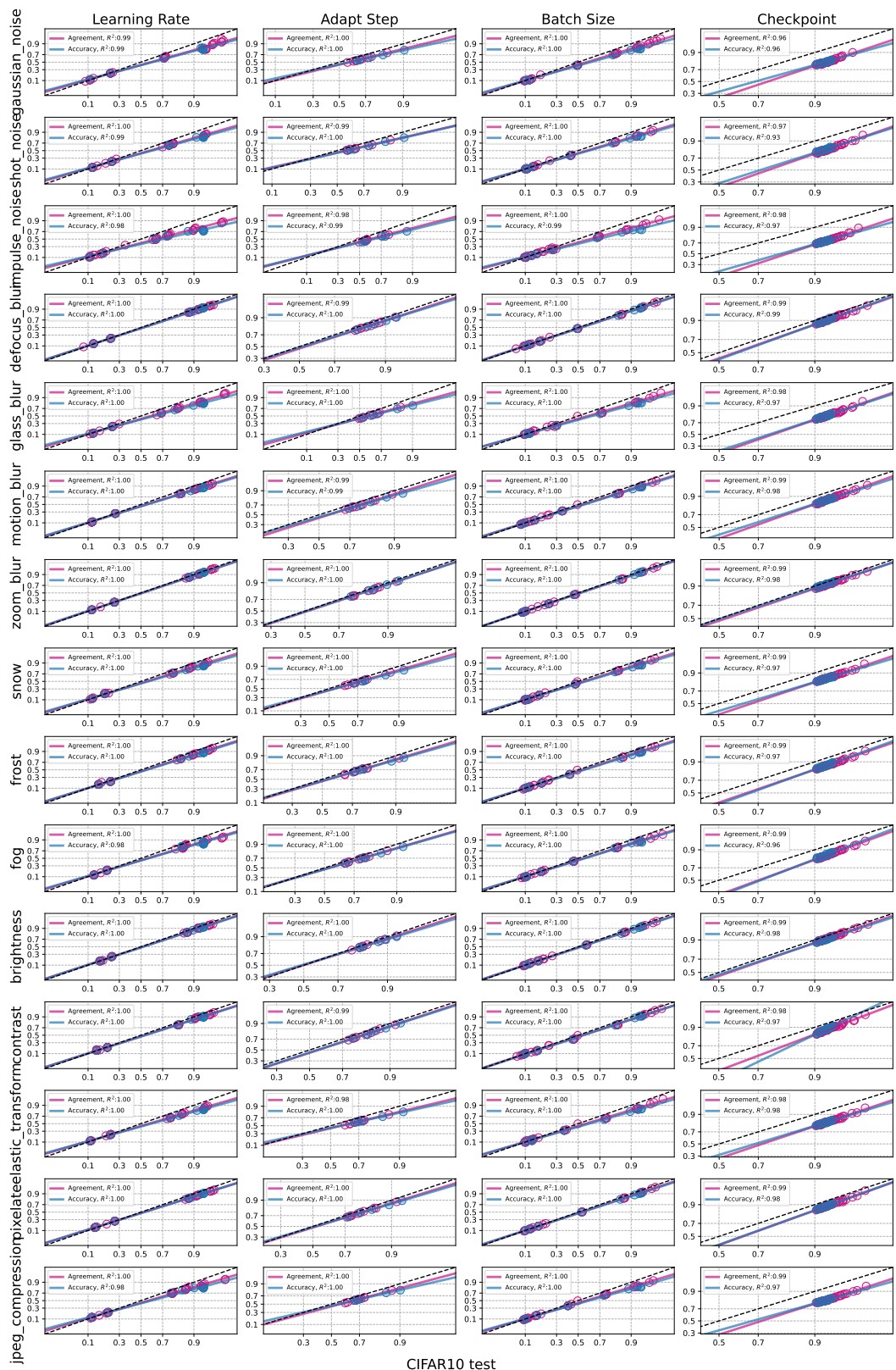

Figure 8: Results on CIFAR10-C corruptions (adaptation hyperparameters of TENT [53]).

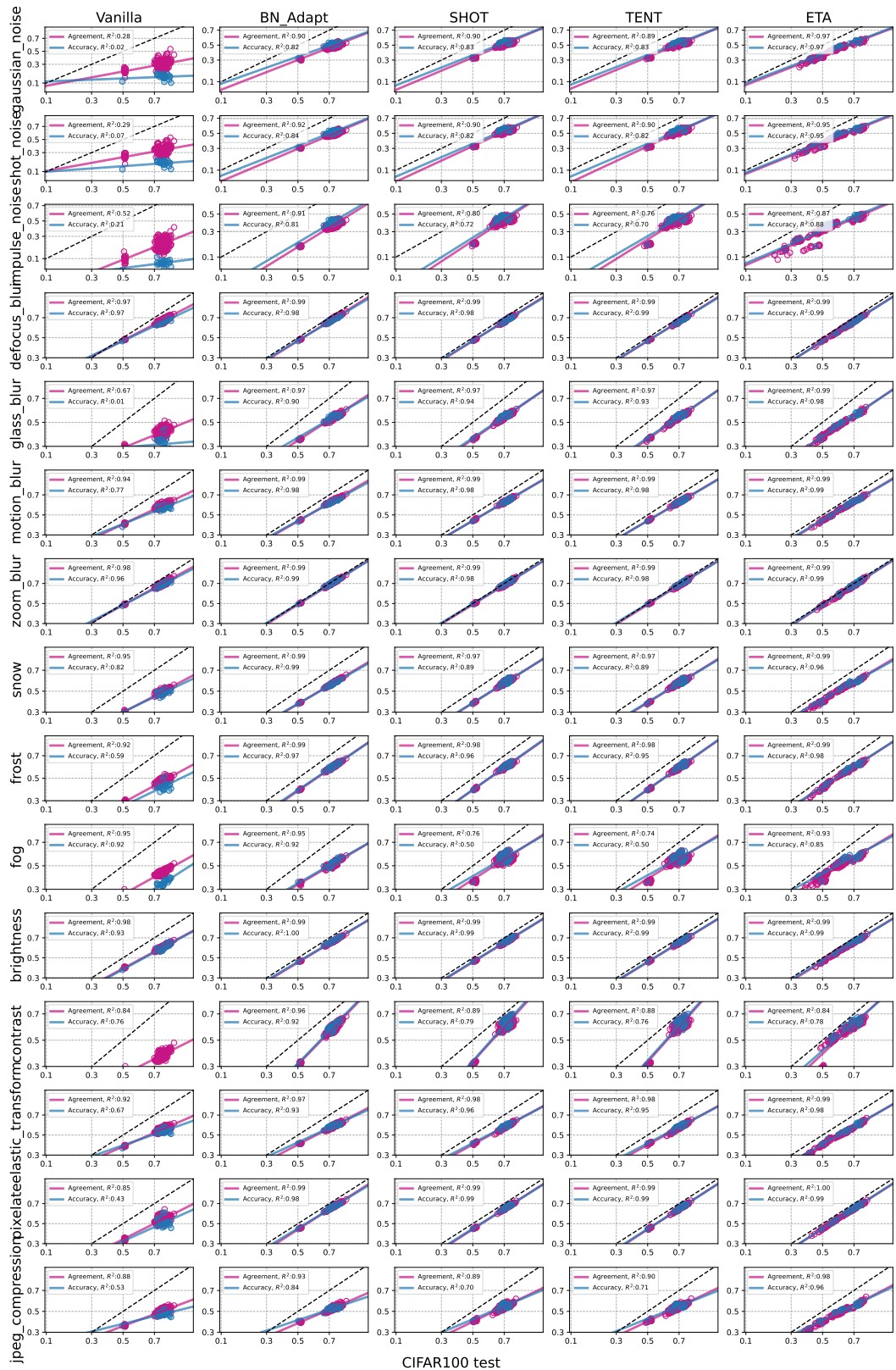

Figure 9: Results on CIFAR100-C corruptions (different architectures), BN_Adapt [44], SHOT [27], TENT [53], and ETA [35].

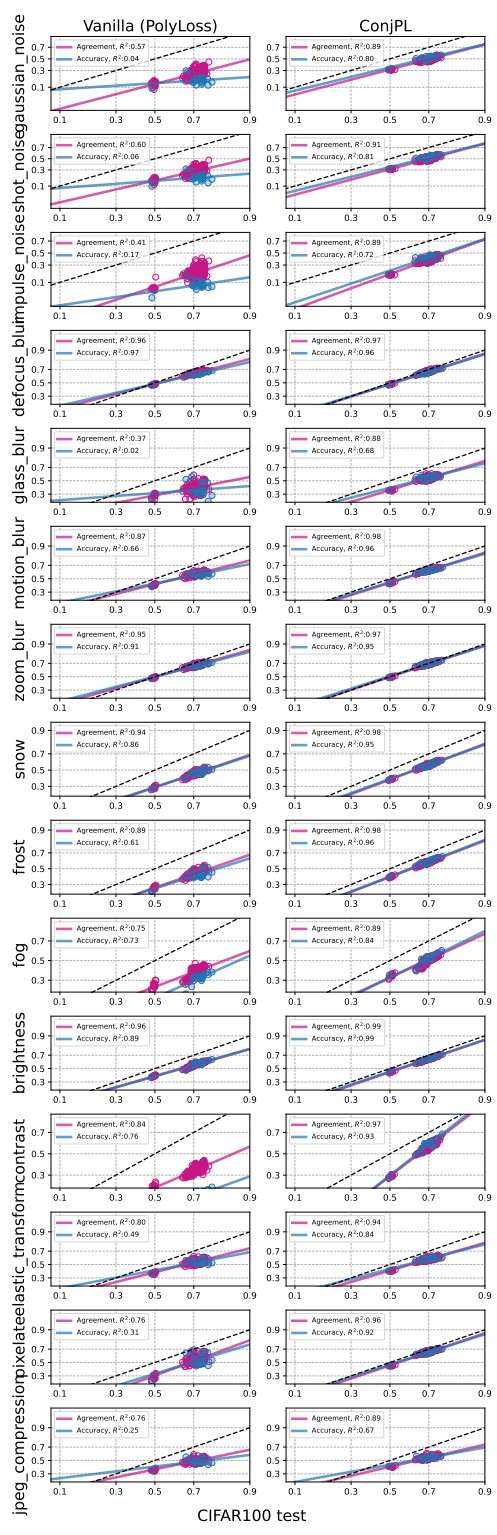

Figure 10: Results on CIFAR100-C corruptions (different architectures), ConjPL [9]

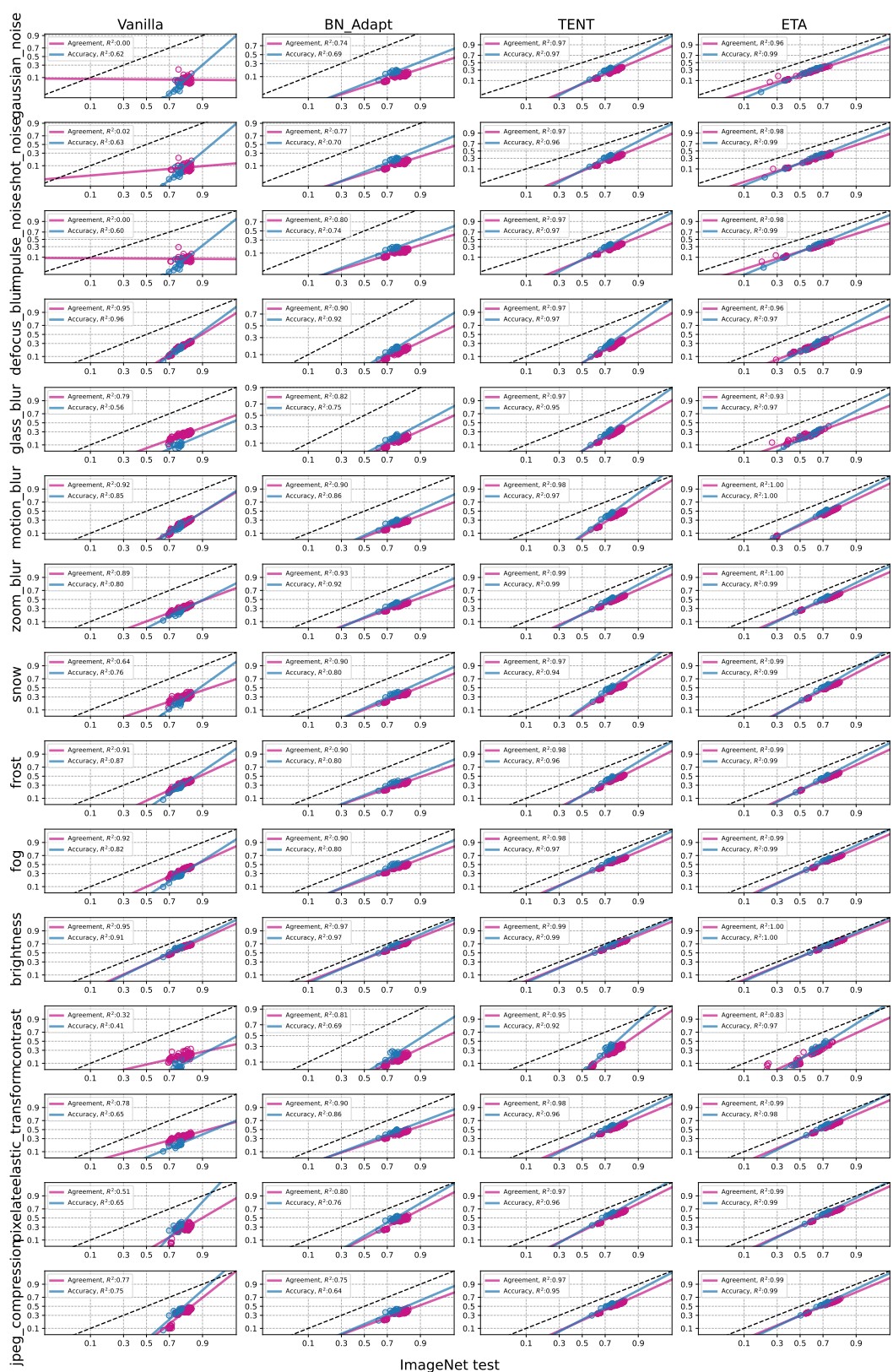

Figure 11: Results on ImageNet-C corruptions (different architectures), BN_Adapt [44], TENT [53], and ETA [35].

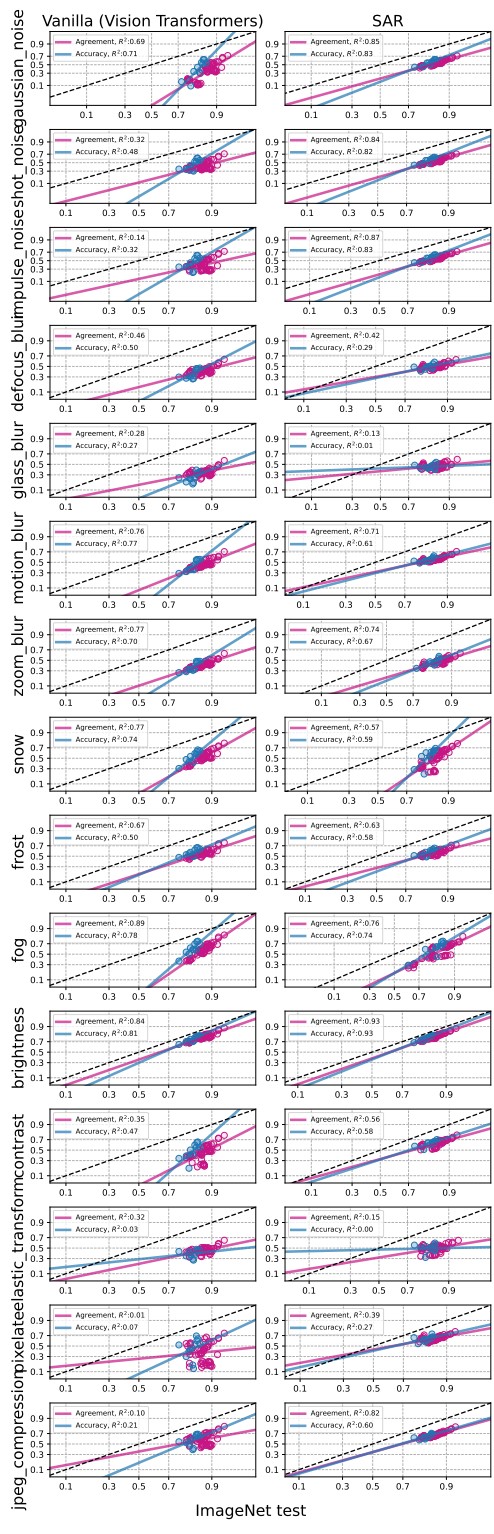

Figure 12: Results on ImageNet-C corruptions (different architectures), SAR [36]

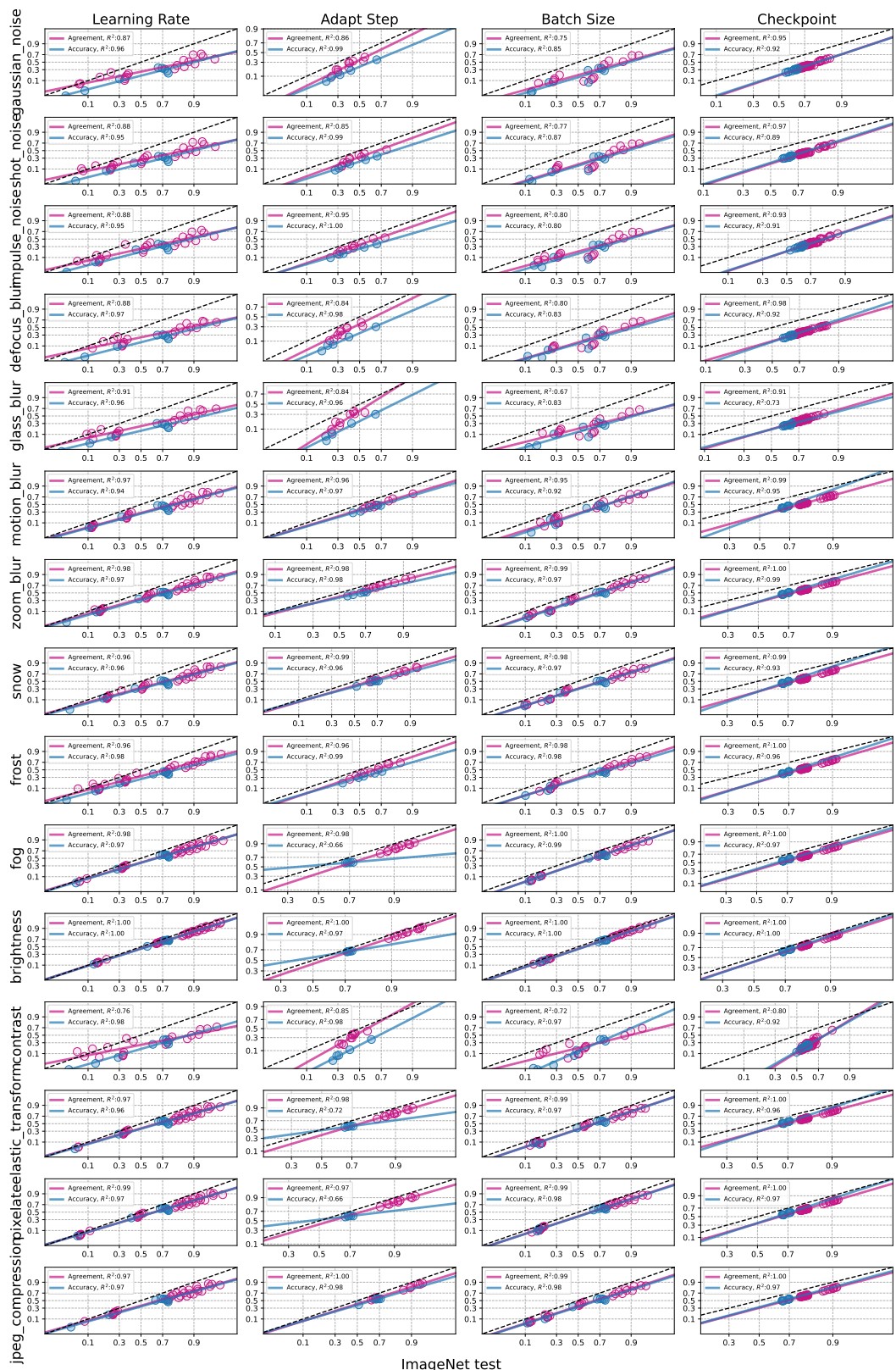

Figure 13: Results on ImageNet-C corruptions (adaptation hyperparameters of ETA [35]).

