# OpenReview forum: "Test-Time Adaptation Induces Stronger Accuracy and Agreement-on-the-Line"
_NeurIPS.cc/2024/Conference — NeurIPS 2024 poster_

### Official Review · Reviewer_1nU3 · 2024-07-02

**Soundness:** 3
**Presentation:** 3
**Contribution:** 3
**Rating:** 6
**Confidence:** 4

**Summary:**

This paper try to achieve reliable TTA by tackling three bottlenecks, including performance evaluation without labeled data, some distribution shifts, and hyperparameter selection for TTA methods.

**Strengths:**

1.	TTA is an important topic, and this work is a relevant and timely contribution.

2.	The experiments are extensive, covering various tasks comprehensively and convincingly.

3.	The presentation is clear and the motivation is well-presented.

**Weaknesses:**

1.	The novelty is relatively limited; it is an application of Baek et al. (2022) to the TTA setup. And the limited theoretical conclusion is from Miller et al. (2019).

2.	It might be necessary to evaluate on more complex tasks, such as object detection and semantic segmentation.

**Questions:**

Overall, the work is an important study of TTA, however there are still some questions.

1.	About Algorithm 1: In TTA, does updating the model parameters require multiple iteration steps? If it does, according to the basic idea of online algorithms, Algorithm 1 does not seem to reflect the model parameters being trained with different batches of data at different time steps. Could you provide a detailed explanation of the difference between the online algorithm you provided and the traditional offline algorithm?

2.	Why the inverse of the cumulative density function of the standard Gaussian distribution is applied on the axes of accuracy and agreement? Is there any intuitions and theoretical explanations?

**Limitations:**

No potential negative societal impact.

---

> ### Author Rebuttal · Authors · 2024-08-07
>
> Dear Reviewer 1nU3,
>
> We greatly appreciate your thoughtful reviews on our paper.
> To address the concerns you raised, we made changes and clarifications as following:
> + Clarified our paper’s novelty,
> + Clarified notations on Algorithm 1,
> + Added theoretical intuition of using inverse of cumulative density function of standard Gaussian distribution.
>
> For each concern, we attempted to address it with individual responses:
>
> **1. The novelty is relatively limited; it is an application of Baek et al.[2] to the TTA setup. And the limited theoretical conclusion is from Miller et al.[1].**
>
> We would like to note that our paper not just applies AGL to TTA setup, but first finds the significant restoration of AGL in various distribution shifts, including those where the linear trends did not hold previously. Previous studies[1-4], including Miller et al.[1] and Baek et al.[2], did not include such restorations, limiting the broader applicability of AGL.
>
> In addition, while Miller et al.[1] only considered the very simple Gaussian setup for the sufficient condition for Theorem 1, our paper presents that TTA involving more complex data with non-linear deep networks satisfy the condition.
>
> **2. It might be necessary to evaluate on more complex tasks, such as object detection and semantic segmentation.**
>
> We appreciate the Reviewer’s suggestion. Since it remains ambiguous to define the agreement between models’ outputs in more than a single label, e.g., object detection and semantic segmentation, our paper mainly focuses on image classification, following Baek et al.[2]. However, extension to such tasks could further extend the understanding of the potential of TTA in stronger linear trends. We will discuss this in our final manuscript.
>
> **3. Algorithm 1 does not seem to reflect the model parameters being trained with different batches of data at different time steps. Could you provide a detailed explanation of the difference between the online algorithm you provided and the traditional offline algorithm?**
>
> We apologize for the confusion regarding Algorithm 1. As mentioned by the reviewer, TTA updates the model parameters in multiple iterations, i.e., at each iteration using the different batch of unlabeled OOD data, as described in L6 of Algorithm 1. We use the notation of x_OOD that means the current batch of OOD test data, which is different at every iteration. We will clarify this by utilizing the different notations for data in different time steps in the final version.
>
> The online algorithm in Algorithm 1 describes the online _test_, which is different from traditional testing. Traditional testing tests the model that is fixed in its pre-trained parameters, while the online "test" is testing a model that is continuously updated its parameters during TTA. In other words, the model tested at iteration of $t-1$ is different from that at iteration $t$. Specifically in Algorithm 1, L6 makes the only difference between traditional and online testing, where L6 updates the model parameters every iteration before testing in L7 and L8.
>
>
> **4. Why the inverse of the cumulative density function of the standard Gaussian distribution is applied on the axes of accuracy and agreement? Is there any intuitions and theoretical explanations?**
>
> Great question! Let us revisit the simple Gaussian distribution setup as described in Eq.2 of our original manuscript, where $Q$ distribution shifts from $P$ only in scale of mean direction and covariance shape: \
> $P(x\mid y)=\mathcal{N}(y\cdot \mu; \Sigma)$, $Q(x\mid y)=\mathcal{N}(y\cdot \alpha\mu;\gamma^2\Sigma)$, \
> where the label $y \in \\{-1,1\\}$ and $\alpha,\gamma$ are constant scalars.
>
> When we consider a linear classifier $f_{\theta}:x \mapsto \theta^{\top}x$, its accuracy on distribution $P$ and $Q$ are defined as \
> $\text{acc}_P(\theta)=\Phi(\frac{\theta^{\top} \mu}{\sqrt{\theta^{\top} \Sigma \theta}})$ ,\
> $\text{acc}_Q(\theta)=\Phi(\frac{\theta^{\top} (\alpha\mu)}{\sqrt{\theta^{\top} \gamma^2\Sigma \theta}})$, \
> where $\Phi$ is the cumulative density function (CDF) of standard Gaussian distribution.
>
> After applying the inverse of $\Phi$ for both accuracies, the linearity between two is calculated by\
> $\frac{\Phi^{-1}(\text{acc}_Q(\theta))}{\Phi^{-1}(\text{acc}_P(\theta))} =  \frac{\theta^{\top} (\alpha\mu)}{\sqrt{\theta^{\top} \gamma^2\Sigma \theta}} / \frac{\theta^{\top} \mu}{\sqrt{\theta^{\top} \Sigma \theta}} =\alpha / \gamma$, \
> which is a constant independent of $\theta$. This means the linear relationship between $\Phi^{-1}(\text{acc}_P(\theta))$ and $\Phi^{-1}(\text{acc}_Q(\theta))$ holds across linear classifiers.
> Therefore, in theory with simple Gaussian setup, such inverse of CDF is required to exactly derive the linear relationship between ID and OOD.
>
> The application of the inverse of the CDF surprisingly improves linear fit beyond simple Gaussian setups, even in more complex real-world distribution shifts, as extensively observed in Miller et al.[1], Baek et al.[2], and our paper.
>
> \
> We hope these responses address the concerns, and let us know if there is any further feedback.
>
> >**References** \
> > [1] Miller et al., Accuracy on the Line: On the Strong Correlation between Out-of-Distribution and In-Distribution Generalization, ICML 2021 \
> [2] Baek et al., Agreement-on-the-Line: Predicting the Performance of Neural Networks under Distribution Shift, NeurIPS 2022 \
> [3] Wenzel et al., Assaying Out-Of-Distribution Generalization in Transfer Learning, NeurIPS 2022 \
> [4] Teney et al., ID and OOD Performance are Sometimes Inversely Correlated on Real-world Datasets, NeurIPS 2023

---

> ### Comment · Reviewer_1nU3 · 2024-08-13
>
> Dear Authors,
>
> Thank you for your response. I maintain my initial positive rating.

---

> > ### Author Response · Authors · 2024-08-13
> >
> > Dear Reviewer 1nU3,
> >
> > Thank you for your positive score, and we are glad that the clarifications were helpful. We will make sure to include such clarifications in our final manuscript.

---

### Official Review · Reviewer_ZucK · 2024-07-09

**Soundness:** 3
**Presentation:** 4
**Contribution:** 2
**Rating:** 7
**Confidence:** 4

**Summary:**

This papers shows that using test-time-adaptation  improve the accuracy of the algorithms used for OOD performance estimation algorithms based on agreement-on-the-line and accuracy-on-the-line. This is shown through a series of experiments. Some justification has also been provided based on prior work,.

**Strengths:**

- OOD performance estimation using unlabeled data is a very important problem and methods based on agreement-on-the-line and accuracy-on-the-line are the state-of-art methods.

- The experimental details are very well presented and the paper is very well written and clear.

**Weaknesses:**

- Intuitively, if we assume that TTA is *perfect*, then we expect ID and OOD accuracy to be the same (i.e., y = x); the model has been fully adapted to the OOD setting and there is no difference between ID and OOD. Also the ID and OOD agreements should intuitively become the same (i.e., y = x). Is this understanding correct? It seems that there is a very basic reason behind the main observation of the paper. ‌Because of this, I think table Table 2 does not show a fair comparison; TTA is making models more similar and shift all the lines towards the y = x line. Is the message of the paper as simple as this? I think this answers the question "What specific mechanisms within these adaptation methods cause the observed phenomena? " asked in the conclusion section.


- In a practice, we use OOD performance estimation with unlabeled data in settings where compute, budget, etc. is limited; thus we cannot collect new labeled data. Given this, is a method that is computationally expensive (that involves test time adaptation) feasible? Also, note that this method does not work if we only have black box access to a model (which is again, typical in potential applications of AGL and ACL).


- What is the condition of theorem 1? It is written that "... correlation if with a bias of zero and a slope ...". Also, the condition in Eq. (2) is very strong. See my comments below for other theoretical results on AGL and ACL that study the problem in other (I believe more realistic) settings where the covariance and mean can shift.


- The literature review is missing many theoretical results on agreement-on-the-line and accuracy-on-the-line. In the past 1-2 years, there has been a lot of theory results on these topics that show that these phenomenon are features of the high-dimensionality.

For example:

[1] N Tripuraneni, B Adlam, J Pennington, Overparameterization improves robustness to covariate shift in high dimensions, NeurIPS 2021.

[2] D Lee, B Moniri, X Huang, E Dobriban, H Hassani, Demystifying Disagreement-on-the-Line in High Dimensions, ICML 2023.

[3] D LeJeune, J Liu, R Heckel, Monotonic Risk Relationships under Distribution Shifts for Regularized Risk Minimization, JMLR, 2024.

[4] A Sanyal, Y Hu, Y Yu, Y Ma, Y Wang, B Schölkopf, Accuracy on the wrong line: On the pitfalls of noisy data for out-of-distribution generalisation, Arxiv 2024.

- The papers above show that these phenomena are fragile; these papers show clear settings where Accuracy-on-the-line and agreement-on-the-line break.

- More important than the previous comment: Lee et al (2023) show that the line for agreement and the line for accuracy can have different intercepts (at least in their toy model). Although the liens have the same slope. This is seen when looking back at the plots of the original real world experiments in Baek et al. This issue needs to be addressed before any real application of AGL for practice. There might be some fundamental issue with the AGL method.

**Questions:**

See weaknesses.

**Limitations:**

The authors should discuss the limitations I mentioned in the weaknesses section.

---

> ### Author Rebuttal · Authors · 2024-08-07
>
> Dear Reviewer ZucK,
>
> We greatly appreciate your thoughtful reviews on our paper. To address the concerns you raised, we made clarifications on:
> + Intuition of linear trend observations,
> + Our study’s contribution under cost-limited scenarios,
> + Condition 1 of the Eq. 2 of our original manuscript,
> + Intuition of condition for ACL guarantee and potential failure of AGL
>
> Detailed responses for each individual concern are as below:
>
> **1. TTA is making models more similar and shift all the lines towards the y = x line. Is the message of the paper as simple as this?**
>
> We would like to clarify that these two phenomena are independent. A substantial gap between ID and OOD accuracies can remain after TTA, but models’ performances may still be strongly correlated. For example, after applying TENT on ImageNet-C Shot Noise, there is still approximately a 40\% gap in ID and OOD accuracy, but strong AGL holds (see submission Fig.1). In other words, strong AGL is not simply explained by the decreased gap between ID and OOD by TTA. In Section 4, we build a theoretical intuition of why TTA induces these strong ACL/AGL trends.
>
> **2. We use OOD performance estimation with unlabeled data in settings where compute, budget, etc. is limited; thus we cannot collect new labeled data. Given this, is a method that is computationally expensive feasible? Also, this method does not work if we only have black box access to a model.**
>
> Great question! TTA is generally very computationally cheap, and arguably much less resource-intensive than collecting labels, which usually requires expensive manual labor. Oftentimes only a very small number of parameters are adapted, making TTA far cheaper than methods like test-time training[5,6] or unsupervised domain adaptation[1-3]] that require a separate pre-training procedure and heavy training. With a small amount of resources, our paper shows that TTA can significantly restore AGL, allowing TTA models to be safely deployed across various distribution shifts. While our method is not feasible with black box access, non-black box settings cover a wide variety of practical scenarios where our method is useful. Moreover, our method often significantly outperforms baselines that do work on black box models with estimation error smaller by a whole factor.
>
> **3. What is the condition of theorem 1?**
>
> We apologize for the confusion. The conditions for Theorem 1 are detailed in Eq. 2. In particular, the class-conditioned distributions must be normally distributed and the distribution shift must only be a simple scale shift, i.e., the mean and covariance can change in magnitude but not their directions. Also, the classifiers $f_{\theta}$ must be linear with no bias. Under these conditions, ID versus OOD accuracy is perfectly linear correlated with a slope of $\alpha / \gamma$.
>
> **3. The condition in Eq. (2) is very strong. Other theoretical results on AGL and ACL that study the problem in other settings where the covariance and mean can shift.**
>
> Thank you for the feedback! We emphasize that, while this assumption seems strong, our empirical study in Section 4 demonstrates that the “scale-shift” condition in Theorem 1 is almost perfectly satisfied via TTA. Indeed, this is precisely why our method induces strong linear trends in practice. Furthermore, the theoretical studies [7,8] that do allow the covariance shape to shift are asymptotic results that rely heavily on the dimensionality of the data and classifiers to go to infinity. They explicitly note that their theory does not guarantee linear trends in finite dimensions (e.g., Figure 2 of [8]). Our paper also include covariance shape shifts such as CIFAR10 vs. CIFAR10-C Gaussian Noise, where classifiers do not exhibit linear trends in finite dimensions.
>
> **4. The literature review is missing many theoretical results on AGL and ACL. The papers above show that these phenomena are fragile. This issue needs to be addressed before any real application of AGL for practice.**
>
> Thank you again for the feedback! We will make sure to include these works in our literature review. We do note that failure modes observed theoretically may not be consistent with common real-world data. For example, Lee et al.[9] show that in random feature linear regression, AGL does not hold in CIFAR10-C. However, in neural networks trained with logistic regression, both our paper and Baek et al. show strong AGL on CIFAR10-C. On the other hand, Sanyal et al.[10] points to AGL breaking under heavy label noise which we don’t observe in many practical settings. Generally, we agree that AGL could be fragile. In fact, our paper directly addresses this problem, expanding, via TTA, the set of distribution shifts where AGL holds. We observe repeatedly across a broad range of distribution shifts that TTA improves ACL and AGL trends, and in all these scenarios the shifts reduce to “scale shifts”  thus satisfying the theoretically sufficient condition for ACL.
> \
> We hope these responses address the concerns, and let us know if there is any further feedback.
>
> >**References** \
> > [1] Ganin et al., Domain Adversarial Training of Neural Networks, JMLR 2016 \
> [2] Sun and Saenko, Deep CORAL: Correlation alignment for deep domain adaptation, ECCV 2016 \
> [3] Li et al., Domain generalization with adversarial feature learning. CVPR 2018 \
> [5] Sun et al., Test-Time Training with Self-Supervision for Generalization under Distribution Shifts, ICML 2020 \
> [6] Gandelsman et al., Test-Time Training with Masked Autoencoders, NeurIPS 2022 \
> [7] Tripuraneni et al., Overparameterization improves robustness to covariate shift in high dimensions, NeurIPS 2021 \
> [8] LeJeune et al., Monotonic Risk Relationships under Distribution Shifts for Regularized Risk Minimization, JMLR 2024 \
> [9] Lee et al., Demystifying Disagreement-on-the-Line in High Dimensions, ICML 2023 \
> [10] Sanyal et al., Accuracy on the wrong line: On the pitfalls of noisy data for out-of-distribution generalisation, Arxiv 2024

---

> ### Comment · Reviewer_ZucK · 2024-08-08
>
> I thank the authors for their detailed response. Below please find my response:
>
> 1. I'm still not convinced that these phenomena are independent. In the same experiment that you mention in the rebuttal, there is a significant improvement in the OOD errors after applying TENT. The OOD and ID accuracies are becoming closer. How do you argue that these phenomena are independent?  There needs to be a systematic experiment that demonstrates that the observed phenomena cannot be simply explained by the decreased gap between ID and OOD by TTA.
>
> 2. The rebuttal resolves my concerns about the computational complexity. Although I believe this that this potential limitation should be further discussed in the paper.
>
> 3. Thanks for the clarification.
>
> 4. Thanks for the clarification. However, this response further strengths my doubt that most probably, the observed phenomena can be simply explained by the decreased gap between ID and OOD by TTA. As you discuss here, after TTA, the covariances and means align very well.
>
> 5. Thanks. I believe such discussion will strengthen the quality of the paper.

---

> > ### Author Response · Authors · 2024-08-09
> > **Further response to Reviewer ZucK's comments**
> >
> > Dear Reviewer Zuck,
> >
> > We really appreciate your quick response to our rebuttal, and we will definitely include further theoretical discussion on ACL/AGL theory and the computational complexity of our method in our final manuscript.
> >
> > You were still concerned that the strong linear trend, which appears after TTA, is caused by model performances approaching the $y=x$ line. We clarify the subtle point below as to **why the closer ID and OOD gap is correlated, but not causal, to the strong linear trend**.
> >
> > As stated in Theorem 1, under distribution shifts that simply “scale” up or down the _norm_ of the mean and covariance, i.e.,  $\mu_{OOD} = \alpha \mu_{ID}$, $\Sigma_{OOD}  = \gamma^2 \Sigma_{ID}$, classifiers can observe perfect linear trends with slope $\alpha / \gamma$. In Section 4, we showed that TTA satisfies this condition by aligning the “direction/shape” of the ID/OOD mean and covariance, while the scaling factor might still be far off, i.e., $\alpha$ << $1$, $\gamma$ >> $1$. Note that the degree of “scale shift” can be large, and in fact, _the perfect trend can lie arbitrarily far from $y=x$_, i.e., $\alpha / \gamma$ << $1$. We see examples of this in our paper, such as TENT on ImageNet-C Shot Noise.
> >
> > On the other hand, any shift, however small, that moves the “direction / shape” of the mean or the covariance matrix breaks the perfect linear trend _in finite dimensions_ (Miller et al.[1]). As an intervention, let us imagine a “Scale TTA” which improves OOD performance by reducing “scales shifts” in covariance in the feature space, but the “shape” of the covariance matrices remain misaligned. We can synthetically simulate the effect of such TTA over toy Gaussian data, where the cosine similarity between the shape of ID and OOD covariance matrices remain less than 1, and $\gamma^2$ decreases from 1 to 0.2 after Scale TTA. Empirically, we can see that the linear trend moves closer to $y=x$ (slope 0.66$\rightarrow$ 0.8), but the strength of the linear trend remains weak ($R^2$ 0.75$\rightarrow$ 0.63). This empirical evidence proves that improving OOD performance does not necessarily make the linear trend stronger.
> >
> > Overall, we see that ACL improves after TTA because TTA aligns the “shape” of the means/covariances, which is the true causal factor for _both_ stronger ACL and the linear trend moving closer to $y = x$. On the other hand, had TTA only reduced “scale shifts” and the “shapes” were unaligned, models would observe weak linear trends even though the slope moves closer to $y=x$.
> >
> > We hope this addresses your concern, and thus makes our paper’s observation and analysis more interesting to you. We will make sure to add the clarification in our final manuscript. Let us know if you have any questions!
> >
> >
> > >**Reference**  \
> > > [1] Miller et al., Accuracy on the Line: On the Strong Correlation between Out-of-Distribution and In-Distribution Generalization, ICML 2021

---

> ### Comment · Reviewer_ZucK · 2024-08-10
>
> I thank the authors for their response. My concerns about the paper are now resolved. However, I believe this discussion is necessary and adding it will significantly increase the quality of the paper. Under the condition that these discussion will be added to the paper, I increase my score to 7 and recommend acceptance.

---

> > ### Author Response · Authors · 2024-08-11
> >
> > Dear Reviewer ZucK,
> >
> > Thank you for your updated score and we are glad that the clarifications were helpful. We will make sure to include the discussion in our final manuscript.

---

### Official Review · Reviewer_JQG9 · 2024-07-12

**Soundness:** 3
**Presentation:** 2
**Contribution:** 3
**Rating:** 5
**Confidence:** 3

**Summary:**

The paper presents a study on how test-time adaptation influences accuracy-on-the-line (ACL) and agreement-on-the-line (AGL). The authors empirically find that TTA methods significantly enhance the ACL and AGL, enabling better OOD performance estimation. Extensive experiments support their findings.

**Strengths:**

1. This paper investigates an under-explored but important observation of TTA.  TTA can boost AGL and ACL phenomenon on out-of-distribution datasets.
2. The observed phenomenon is verified under diverse settings, including learning rates, the number of adapting steps, batch sizes, and the early-stopped epoch for pretraining, etc.

**Weaknesses:**

1. The scope should be revised. TTA encompasses more than just feature extractor adaptation, but also classifier adaptation such as T3A [1]. Additionally, it appears that the author does not employ TTA methods with a memory bank [1-4]. Therefore, the paper should clearly specify which types of TTA methods support the observations.
2. The experimental setting is idealized. It seems that the paper only focuses that how TTA always improves accuracy. However, TTA might fail under continuously changing distributions, especially in cases of small batch sizes (e.g.,1), or significant distribution shifts. The author should provide a discussion of these scenarios.
3. The analysis in section 4.2 is not supervising, since BN_Adapt and TENT only modify the $\gamma$ and $\beta$ in BN layers to make the output features similar to the target domain. It would be better to discuss whether other non-BN methods align with these results.
4. One key advantage of TTA is that it does not require source domain data. However, the AGL-based method requires it, which limits the contribution of this paper.

[1] Iwasawa Y, Matsuo Y. Test-time classifier adjustment module for model-agnostic domain generalization. In NeurIPS. 2021.

[2] Jang M, Chung S Y, Chung H W. Test-time adaptation via self-training with nearest neighbor information. In ICLR. 2023.

[3] Yuan L, Xie B, Li S. Robust test-time adaptation in dynamic scenarios. In CVPR. 2023.

[4] Wang S, Zhang D, Yan Z, et al. Feature alignment and uniformity for test time adaptation. IN CVPR. 2023.

**Questions:**

1. Visualizing the change of correlation coefficient as the model adapts to the test data flow could be more beneficial for understanding the contribution of the paper.

---

> ### Author Rebuttal · Authors · 2024-08-07
>
> Dear Reviewer JQG9,
>
> We greatly appreciate your thoughtful reviews on our paper. To address the concerns you raised, we included additional results and clarifications as following:
> + Added T3A as non-BN method and its AGL visualizations and feature alignment analysis,
> + Clarified our paper’s results on when TTA does not improve accuracy,
> + Clarified our method’s robustness under ID-data limited setup,
> + Added visualizations of coefficient correlations during the progress of TTA.
>
> Detailed responses for each individual concern are as below:
>
> **1, 3. TTA encompasses more than just feature extractor adaptation, but also classifier adaptation such as T3A. Also, it would be better to discuss whether other non-BN methods align with these results.**
>
> We appreciate the Reviewer's feedback! To address the concern, in rebuttal Figure 2  and Table 2, we include
> + T3A[1]’s AGL visualizations as well as feature alignment analysis on CIFAR10-C Gaussian Noise.
>
> Since T3A[1] is applied to non-BN networks, we train the networks without BN layers and apply T3A. In Fig.2, we similarly observe that, using T3A (left) to adapt the last classifier only results in stronger ACL compared to models before adaptation (labeled as “Vanilla w/o BN”) (right). Specifically, the correlation coefficient of ACL increases from 0.29 to 0.80. Still, T3A’s correlation coefficients in both accuracy and agreement are relatively weak compared to BN-based methods (e.g., 0.80 and 0.46 in T3A vs. 1.0 and 1.0 in TENT). We also make note that T3A’s accuracy and agreement lines have different biases.
>
> However, while T3A does improve ACL trends, it does not seem to satisfy the same sufficient theoretical condition of ACL that BN-adaptation methods satisfy. As T3A only modifies the final classifier, the distribution shift from ID to OOD in the penultimate representation space remains more complicated than a simple scale shift. In Table 2 we reported the cosine similarity between the ID and OOD mean and covariance of Vanilla w/o BN, which is the same as that of T3A, since no updates in feature extractor. The results in Table 2 show that their cosine similarity are approximately 0.78 (averaged over different architectures), with standard deviation exceeding 0.1. The cosine similarities are _much lower compared to those of BN-based TTA methods_ (e.g, TENT has 0.990$\pm$0.005 and 0.974$\pm$0.011 for mean and covariance).
>
> Overall, our method may also apply to non-BN adaptation methods such as T3A, but the theory in our work does not explain why stronger ACL trends appear in this setting. This is an important open question we hope to understand further in the future. Due to the limited timeline of rebuttal, we could not examine other baselines mentioned by the Reviewer.
>
> **2. The paper only focuses that how TTA always improves accuracy, but TTA might fail under continuously changing distributions, especially in cases of small batch sizes (e.g.,1), or significant distribution shifts.**
>
> We would like to clarify that our paper already explores scenarios where the TTA degrades OOD performance, including:
>
>  * Poor hyperparameter choices that lead to low ID and OOD accuracy, such as extremely small batch sizes (e.g., 1 on CIFAR10-C). However, these models still  conform to the same linear ACL/AGL trends (Figs. 2 and 7 of original submission). In fact, our theoretical explanation generalizes to such settings. In Table 2 of original submission, we show that the features after TENT with batch size of 1 also demonstrate near perfect alignment (shown by near-zero standard deviation).
>
> * Real-world distribution shifts, e.g., CIFAR10.1, ImageNet-V2, FMoW-WILDS, where TTA hurts performances, but maintains linear trends (Fig. 3  of original submission).
>
> Importantly, _AGL is restored regardless of whether TTA succeeds_ for any one hyperparameter choice $-$ this is precisely why we can employ our method for hyperparameter optimization.
>
> **4. One key advantage of TTA is that it does not require source domain data. However, the AGL-based method requires it, which limits the contribution of this paper.**
>
> Our method works well at estimating OOD accuracy given just a small number of ID data (i.e., 1-5% of total), as demonstrated in Section D of our submission. In fact, even with limited data, our methods outperform other baselines, e.g., ATC and DoC-feat, which also require the access to ID data for temperature scaling.
>
> The fact of the matter is even recent TTA methods are highly sensitive to hyperparameters and their OOD is often unpredictable. Zhao et al.[2] recently pointed out this issue. Using a reasonably small amount of ID data, which is relatively easy-to-obtain than labeled OOD data, our method can greatly enhance the reliability of TTA methods in their practical usage.
>
> **5. Visualizing the change of correlation coefficient as the model adapts to the test data flow could be more beneficial for understanding the contribution of the paper.**
>
> Great suggestions! In Figure 3 of our rebuttal PDF we’ve added a
> * Visualization of BN_Adapt, TENT, and ETA's correlation coefficient ($R^2$) of ACL as TTA progresses on CIFAR10-C Gaussian Noise and CIFAR100-C Glass Blur
>
> For comparison, we also added the correlation coefficient of the Vanilla model at iteration=0.
>
> We observed that the strong correlation rapidly appears (e.g., 0.4 to 1.0 in TENT on CIFAR10-C Gaussian Noise) at the very beginning of adaptation (i.e., iteration=1) and remains high until the end of adaptation. Our finding shows that strong AGL induced by TTA appears in the very early stage of adaptation, and remains strong over time.
>
> \
> We hope these responses address the concerns, and let us know if there is any further feedback.
>
>
> >**References** \
> > [1] Iwasawa et al., Test-time classifier adjustment module for model-agnostic domain generalization. In NeurIPS 2021 \
> [2] Zhao et al., On pitfalls of test-time adaptation, ICML 2023

---

> > ### Comment · Reviewer_JQG9 · 2024-08-11
> >
> > Dear Authors,
> >
> > Thank you for your response. I maintain my positive rating.
> >
> > Best,
> >
> > Reviewer JQG9

---

> ### Author Response · Authors · 2024-08-11
>
> Dear Reviewer JQG9,
>
> Thank you for your positive score, and we are glad that the clarifications were helpful. We will make sure to include such clarifications in our final manuscript.

---

### Official Review · Reviewer_puv1 · 2024-07-13

**Soundness:** 3
**Presentation:** 3
**Contribution:** 2
**Rating:** 5
**Confidence:** 2

**Summary:**

This paper presents observations that TTA models exhibit strong agreement-on-the-line (AGL) and accuracy-on-the-line (ACL) phenomenon, which persists across a wide range of distribution shifts and models. Leveraging this observation, the authors apply methods to estimate OOD accuracy without labeled data and perform a hyperparameter selection task for a TTA model. The proposed methods are evaluated extensively, demonstrating their effectiveness in TTA settings.

**Strengths:**

1. The identification of the AGL phenomenon in TTA models is interesting and novel. This observation is leveraged effectively to address significant challenges in the field, including performance evaluation and hyperparameter tuning.

2. The authors provide extensive experimental results to support their claims. The evaluation covers various TTA methods and datasets, demonstrating the robustness and generality of the proposed approach.

**Weaknesses:**

1. The paper lacks a comprehensive comparison with the extensive body of literature on model selection methods under distribution shifts (e.g., i-iv below) that share a same goal of predicting OOD performances and thereby achieving a better model selection performance.

2. While the paper examines the hyperparameter selection task for individual TTA methods, a more interesting and practical task would be to choose the best model among various TTA methods. The suggested metric might have a slope unique to each TTA method, potentially complicating the model selection task among different TTA methods. Addressing this issue would enhance the practical applicability of the proposed approach.

3. The paper is missing a core theoretical explanation of how TTA methods can result in a feature space that satisfies the sufficient condition in Theorem 1. Without this theoretical foundation, I have an impression that this paper could be interpreted as a simple application of Baek et al. and Miller et al.'s findings to the TTA setting. Providing a deeper theoretical insight would strengthen the contribution of the paper.

i. Hu, D., Liang, J., Liew, J. H., Xue, C., Bai, S., & Wang, X. (2024). Mixed Samples as Probes for Unsupervised Model Selection in Domain Adaptation. In NeurIPS.

ii. Musgrave, K., Belongie, S., & Lim, S. N. (2022). Benchmarking validation methods for unsupervised domain adaptation. arXiv preprint arXiv:2208.07360, 2(6), 12.

iii. Yang, J., Qian, H., Xu, Y., & Xie, L. (2023). Can we evaluate domain adaptation models without target-domain labels? a metric for unsupervised evaluation of domain adaptation. arXiv preprint arXiv:2305.18712.

iv. Saito, K., Kim, D., Teterwak, P., Sclaroff, S., Darrell, T., & Saenko, K. (2021). Tune it the right way: Unsupervised validation of domain adaptation via soft neighborhood density. In ICCV.

**Questions:**

Kindly see the weaknesses part.

**Limitations:**

The authors partially discuss the limitations in the paper.

---

> ### Author Rebuttal · Authors · 2024-08-07
>
> Dear Reviewer puv1,
>
> We greatly appreciate your thoughtful reviews on our paper. To address the concerns you raised, we included additional results and clarifications as following:
> + Comparisons with five existing model selection baselines,
> + Best TTA methods selection results using our method,
> + Clarified our paper’s novelty and insight provided by Section 4 analysis.
>
> We address the individual concern as below:
>
> **1. The paper lacks a comprehensive comparison with the extensive body of literature on model selection methods under distribution shifts.**
>
> To address the concern we add comparisons in Table 1 of the rebuttal pdf as below:
> + Hyperparameter selection with MixVal[1], Entropy[2], IM[3], Corr-C[4], and SND[5] on CIFAR10-C over all corruptions, ImageNet-R, and Camelyon17-WILDS.
>
> We evaluate methods across various hyperparameters including architecture, learning rates, early-stopped checkpoints, batch sizes, and adaptation steps. Our method consistently outperforms or is competitive against other baselines across datasets and hyperparameters, resulting in state-of-the-art performances on average. We noticed that current state-of-the-art UDA model selection methods, i.e., MixVal[1] or IM[3], perform well on CIFAR10-C and ImageNet-R, but they critically fail in Camelyon17-WILDS, e.g., validation error of 7.98\% in MixVal and 23.52\% in IM.
>
> In contrast, our method is consistently reliable across shifts, including those where other baselines fail, e.g., 0.62\% in Camelyon17-WILDS. The failure modes of existing baselines might come from their assumptions, e.g., low-density separation, that do not generalize to such distribution shifts. Overall, our method of inducing AGL using TTA is most reliable for model selection.
>
> **2. A more interesting and practical task would be to choose the best model among various TTA methods.**
>
> Great suggestions! Our method can be easily applied to select the best TTA strategy. Our unsupervised hyperparameter selection strategy not only effectively chooses the optimal hyperparameters for each TTA method, but _using AGL-based estimators, can also closely predict the OOD accuracy of each method + hyperparameter choice pair_. These estimates can be used to select the overall best strategy.
>
> In Fig.1 of the rebuttal pdf, we present a
> + Comparison of our OOD accuracy estimates in SHOT, BN, TENT, ConjPL, and ETA on CIFAR10-C over all corruptions
>
> For each TTA method, we plot the true (GT) OOD accuracy on the x-axis, and our estimates on the y-axis. Plotted as “x” marks are TTA strategies paired with the optimal hyperparameters selected using our method. For each TTA strategy, we also report its OOD accuracy averaged over all hyperparameter choices and our average estimates of these accuracies, marked as “o”.
>
> You can see that our method precisely estimates both the best and the averaged OOD performances of each TTA strategy (i.e., very close to $y=x$ line). Notably, our estimates preserve the ranking of the TTA methods by their OOD accuracy, i.e., almost no reversed order in $y$-axis compared to $x$-axis. This indicates that our methods can be easily adapted for selecting the best TTA models overall.
>
> **3. Without this theoretical foundation, I have an impression that this paper could be interpreted as a simple application of Baek et al. and Miller et al.'s findings to the TTA setting.**
>
> Thank you for the feedback! Beyond our methodological contributions, we identify an interesting behavior where TTA methods reduce distribution shift in each class distribution to just a scale shift in the representation space. This happens to be the theoretically sufficient condition for ACL as described in Miller et al [6] and our Theorem 1. We take this even further in Table 1 of our original submission, where we show that if we measure the scale shift in the representation space and compute the theoretical slope, it closely matches the actual empirical slope of the ACL trend. We are the first to characterize this behavior of TTA methods and this is a novel contribution, on its own, unexplored by Miller et al. [6] or Baek et al. [7].
>
> We provide rough intuition for why this may happen in BN_Adapt. In BN_Adapt, the data is standardized in each BN layer using the OOD 1st and 2nd moments calculated at test time instead of the ID statistics saved during training. Imagine that the final-layer features are the output of a BN layer. Before TTA, due to frozen BN stats, the ID features are roughly standardized to mean 0 and unit variance, while the OOD features have shifted mean and covariance. Then after TTA, the test-time BN stats also standardizes the OOD features closer to mean 0 and unit variance. However, while BN_Adapt standardizes the _overall_ OOD distribution, BN may not be able to deal with any scale shifts within each _class-conditioned_ distribution. We can follow-up during the discussion period with a more precisely constructed argument if interested.
>
> We hope these responses address the concerns, and let us know if there is any further feedback.
>
>
> >**References** \
> > [1] Hu et al., Mixed Samples as Probes for Unsupervised Model Selection in Domain Adaptation, NeurIPS 2023 \
> [2] Morerio et al., Minimal-entropy correlation alignment for unsupervised deep domain adaptation, ICLR 2018 \
> [3] Musgrave et al., Benchmarking validation methods for unsupervised domain adaptation, arXiv 2022 \
> [4] Tu et al., Assessing model out-of-distribution generalization with softmax prediction probability baselines and a correlation method, 2023 \
> [5] Saito et al., Tune it the right way: Unsupervised validation of domain adaptation via soft neighborhood density,  ICCV 2021 \
> [6] Miller et al., Accuracy on the Line: On the Strong Correlation between Out-of-Distribution and In-Distribution Generalization, ICML 2021 \
> [7] Baek et al., Agreement-on-the-Line: Predicting the Performance of Neural Networks under Distribution Shift, NeurIPS 2022

---

> > ### Comment · Reviewer_puv1 · 2024-08-13
> >
> > I appreciate the authors' efforts in addressing my questions and conducting additional experiments. I believe the newly added experiments enhance the practical value of the findings presented in the paper. Considering the rebuttal and the comments from other reviewers, I have increased my score to 5.

---

> ### Author Response · Authors · 2024-08-13
>
> Dear Reviewer puv1,
>
> Thank you once again for your thoughtful and detailed review. We kindly remind you of our rebuttal that include:
> * (**W1**) Adding a comparison to existing model selection baselines,
> * (**W2**) Demonstrating best TTA method selection,
> * (**W3**) Providing intuition on how TTA satisfies the condition in Theorem 1, and clarified novelty against Miller et al. and Baek et al.,
>
> to address your concerns. We are happy to discuss with you further if you have any other questions or feedback!

---

> ### Author Response · Authors · 2024-08-14
>
> Dear Reviewer puv1,
>
> Thank you for updating your score, and we are glad that the clarifications were helpful. We will make sure to include such results and clarifications in our final manuscript.

---

### Author Rebuttal · Authors · 2024-08-07

We greatly appreciate all four reviewers' valuable feedback and thoughtful suggestions.

The reviewers highlighted the following strengths of our paper:

* Our paper investigates the under-explored but novel and important observation of TTA inducing AGL phenomenon (Reviewer puv1, Reviewer JQG9).
* Our paper addresses significant and important challenges, such as OOD performance estimation and hyperparameter tuning, and demonstrates its effectiveness (Reviewer puv1, Reviewer ZucK, Reviewer 1nU3).
* Our paper includes extensive experimental results to support our claims (Reviewer puv1, Reviewer JQG9, Reviewer 1nU3).

Overall summary of additional experiments, clarifications, and discussions are as below:
* We added clarifications on our paper's novelty compared to existing TTA studies and Miller et al.[1] and Baek et al.[2] (Reviewer puv1, Reviewer 1nU3)
* We additionally tested our method for comparison with other model selection baselines and best TTA methods selection application (Reviewer puv1)
* We discussed our theoretical insight on how TTA achieves features that satisfy the theoretical guarantee of ACL (Reviewer puv1)
* We included additional analysis on how TTA induces strong AGL, by adding T3A as new TTA baseline and tracking correlation coefficient during TTA. (Reviewer JQG9)
* We added clarifications on our observations on circumstances including those TTA fails or access to ID data is limited (Reviewer JQG9)
* We added clarifications on why we observe stronger AGL in TTA is interesting, and TTA's feasibility under compute-limited scenario for better OOD estimation (Reviewer ZucK)
* We discussed on our paper compared to existing theoretical studies on ACL and AGL (Reviewer ZucK)
* We added clarification on lack of details, including condition for theoretical setup in Theorem 1, concept of online test in Algorithm 1, and theoretical insight in using inverse of CDF of standard Gaussian distribution for better linear fit (Reviewer ZucK, Reviewer 1nU3)

To address the concerns and suggestions raised by the reviewers, we uploaded additional pdf that includes figures and tables.

---

### Decision · Program_Chairs · 2024-09-25

**Decision:**

Accept (poster)

**Comment:**

This paper empirically observes that test-time adaptation methods strengthen the "accuracy-on-the-line" and "agreement-on-the-line" phenomena. All reviewers were in agreement that the paper is worth publishing. However, they also raised some concerns whose resolutions  should be in in the final version (e.g., adding the experiments on model selection).